# STAR NDSI collection: A cloud-free MODIS NDSI dataset (2001–2020) for China

Yinghong Jing[1], Xinghua Li[2], and Huanfeng Shen[1,3]

[1]School of Resource and Environmental Sciences, Wuhan University, Wuhan 430079, China
5  [2]School of Remote Sensing and Information Engineering, Wuhan University, Wuhan 430079, China
[3]Collaborative Innovation Centre of Geospatial Technology, Wuhan 430079, China

*Correspondence to:* Xinghua Li (lixinghua5540@whu.edu.cn), Huanfeng Shen (shenhf@whu.edu.cn)

**Abstract.** Snow dynamics are crucial in ecosystems, affecting radiation balance, hydrological cycles, biodiversity, and human activities. Snow areas with notably diverse characteristics are extensively distributed in China, mainly including Northern Xinjiang (NX), Northeast China (NC), and Qinghai-Tibet Plateau (QTP). Spatio-temporal continuous snow monitoring is indispensable for ecosystem maintenance. Nevertheless, the formidable challenge of cloud obscuration severely impedes data collection. In the past decades, abundant binary snow cover area (SCA) maps have been retrieved from moderate resolution imaging spectroradiometer (MODIS) datasets. However, the integrated normalized difference snow index (NDSI) maps containing additional details on snow cover extent are still extremely scarce. In this study, a recent 20-year stretch seamless Terra–Aqua MODIS NDSI collection in China is generated using a Spatio-Temporal Adaptive fusion method with erroR correction (STAR), which comprehensively considers spatial and temporal contextual information. Evaluation tests confirm that the cloud-free STAR NDSI collection is superior to two baseline datasets. The omission error decreased by 10% in NX compared to the snow cover extent product, and the average correlation coefficient increased by 0.11 compared to the global cloud-gap-filled MODIS NDSI product. Consequently, this collection can serve as a basic dataset for hydrological and climatic modeling to explore various critical environmental issues in China. This collection is available from https://doi.org/10.5281/zenodo.5644386 (Jing et al., 2021).

## 1 Introduction

Snow is a fundamental component of the cryosphere, strongly interacting with global energy budgets and hydrological dynamics (Hall et al., 1995). Snow cover has a remarkable impact on the Earth's radiation balance due to its highly reflective nature, thus generating feedback in the global climate system (Konzelmann and Ohmura, 1995). Up to one-sixth of the world's population relies on meltwater from glaciers and snowpacks for drinking, irrigation, hydropower generation, and industrial production (Barnett et al., 2005). Therefore, snow dynamics have a profound impact on climate change and human activities. The snow cover extent of the northern hemisphere has continued to decrease since the mid-20th century (Pachauri and Meyer, 2014). However, regional-scale snow variations in different environmental conditions present mixed trends due to the strong sensitivity of snow cover to climate change (Bormann et al., 2018). The snow cover regions in China are extensively distributed

with remarkable spatial and temporal heterogeneity (Wang et al., 2018), mainly in Northern Xinjiang (NX), Northeast China (NC), and Qinghai-Tibet Plateau (QTP). Therefore, accurate snow cover acquisition in China is significant for snow pattern analysis, water resource management, climate change monitoring, etc.

China has conducted large-scale observations of snow parameters since the 1950s through meteorological stations, providing a valuable data basis for long-term snow-related studies. However, accurately depicting the snow characteristics in China, especially on QTP dominated by patchy and shallow snow, is difficult due to the sparsely and unevenly distributed traditional in-situ observations. Satellite-based remote sensing is a prominent alternative for continuous snow cover monitoring at meso and macro scales. Moderate resolution imaging spectroradiometer (MODIS) snow cover datasets are extensively used for various hydrological and climatological applications due to their relatively high spatial and temporal resolutions. At present, the Collection 5 (C5) suite providing snow cover area (SCA) and fractional snow cover (FSC) data, and the Collection 6 (C6) suite providing normalized difference snow index (NDSI) data are the most appealing representatives (Riggs and Hall, 2015). However, the main constraint of optical remote-sensed datasets, including MODIS C5 and C6 snow cover datasets, is cloud persistence.

Numerous algorithms have been proposed in the past decades to improve the spatio-temporal continuity of MODIS C5 snow cover datasets. Cloud removal algorithms can be categorized into single-source feature fusion methods and multi-source data fusion methods considering data sources. Single-source feature fusion methods fill the gaps based on homologous contextual information, relying on the spatio-temporal correlations of snow features. These methods have evolved from the classical Terra and Aqua combination (TAC; Parajka and Bloschl, 2008), multi-day combination (MDC; Gafurov and Bardossy, 2009), and snow-line method (SNOWL; Parajka et al., 2010) to complex spatio-temporal union methods. For example, Gafurov et al. (2015) proposed a four-step method to generate cloud-free MODIS SCA maps, successively combining TAC, neighborhood filtering, MDC, and classification tree. Dariane et al. (2017) suggested the aggregation of TAC, MDC, SNOWL, and neighborhood filtering with elevation constraints to fill the cloud-covered gaps. Li et al. (2017) developed an adaptive spatio-temporal weighted method to reclassify the cloudy pixels. These methods for binary SCA mapping have achieved satisfactory cloud removal effectiveness and accuracy. Multi-source data fusion methods (Akyurek et al., 2010; Brown et al., 2010; Chen et al., 2018; Gafurov et al., 2015; Gao et al., 2011; He et al., 2017) maximize the complementarity among heterogeneous datasets from optical, microwave, and/or station measurements. This type of method is effective for filling the continuous gaps in space and time when the supplementary data are of high quality in the cloud-obscured regions (Li et al., 2019). In addition to traditional methods, learning-based methods are increasingly applied to snow cover mapping due to their satisfactory capabilities for nonlinear expressiveness (Yuan et al., 2020). SCA and FSC maps can be generated by exploring the relationship between snow coverage and MODIS reflectances combined with ancillary factors, including NDSI, temperature, vegetation, and terrain parameters. As a representative of learning-based methods, artificial neural networks have been successfully utilized to model the relationship (Dobreva and Klein, 2011; Hou and Huang, 2014; Moosavi et al., 2014;

Çiftçi et al., 2017; Kuter, 2021). Such methods are relatively uncertain but promising because the accuracy substantially relies on the quantity and quality of training data.

Increasing studies have moved to the MODIS C6 suite since its release in 2016. In C6 data, snow cover is reported as NDSI rather than SCA and FSC. NDSI is an index that is related to the snow presence in a pixel and is a more accurate description of snow fraction as compared to SCA and FSC (Riggs and Hall, 2015; Riggs et al., 2017). The clear-sky accuracy of C6 NDSI datasets is robust compared to higher resolution remote-sensed images (such as Landsat and Sentinel series) and in-situ measurements (Crawford, 2015; Zhang et al., 2019; Aalstad et al., 2020). As a basic data, it has the significant advantage

of allowing users to more accurately determine SCA or FSC for their particular study areas and application requirements (Hall et al., 2019). For example, several optimal classification thresholds for SCA (Huang et al., 2018; Malmros et al., 2018; Tong et al., 2020) and specially tuned mapping methods for FSC (Kuter et al., 2018; Hou et al., 2020; Zhang et al., 2021) were designed to generate regional SCA and FSC datasets from NDSI snow cover datasets, which were superior to the globally harmonized algorithms in C5 data. However, severe cloud contamination also limits the application of NDSI datasets, resulting

in many studies only considering cloud-free areas (Kuter et al., 2018; Malmros et al., 2018; Tong et al., 2020; Hou et al., 2020; Zhang et al., 2021). Therefore, a cloud-free NDSI dataset is significant for in-depth snow cover research. Since the aforementioned cloud removal methods were generally designed for binary SCA, their applicability to NDSI with more complicated spatio-temporal characteristics should be improved. Thus, several gap-filling methods with an associated concern of spatial and temporal correlations of snow presence were proposed to remove clouds from NDSI (Jing et al., 2019; Chen et

al., 2020; Li et al., 2020). Among these methods, the spatio-temporal feature-based methods with relatively high robustness are more effective for improving NDSI datasets (Jing et al., 2019).

Many studies on snow monitoring in China are available, and most of these studies focus on binary SCA mapping. On the regional scale, QTP, which is known as the world's third pole, plays a key role in the global climate system. Nevertheless, snow cover mapping is particularly challenging over QTP due to the frequent cloud cover resembling fragmented snow. A

large number of studies have demonstrated that the snow cover variability over QTP is extremely complex, with significant spatio-temporal heterogeneity (Gao et al., 2012; Tang et al., 2013; Yu et al., 2016; Liang et al., 2017; Zhang et al., 2012). NX (Wang et al., 2008) and NC (Che et al., 2016) located in mid-latitude areas are dominated by seasonal snow cover. Che et al. (2019) presented an integrated snow cover dataset from a distributed hydrometeorological observation network in the Heihe River Basin, which achieved a prominent demonstration of data synthesis at a watershed scale. In addition, the large-scale

transient snow cover areas increase the level of challenge for generating high-quality snow cover datasets. On the national scale, Huang et al. (2016) obtained a long-term cloud-removed SCA product using a multi-source data fusion method. Despite many relevant studies, only a few cloud-free snow cover datasets have been released publicly.

Several typical long-term cloud-free snow cover products available online are listed in Table 1 (datasets are referenced via DOI), which cover most snow-dominated regions in China. Huang (2020) provided MODIS daily cloudless SCA products

with relatively accurate snow detection capabilities in Northern Hemisphere based on multi-source data. Muhammad and Thapa (2020, 2021) obtained eight-day/daily MODIS SCA and glacier composite datasets for High Mountain Asia by aggregating seasonal, temporal, and spatial filters, which can serve as a valuable input for hydrological and glaciological investigations. Hao et al. (2021; 2022) yielded two long-term daily SCA datasets over China through a series of processes such as quality control, cloud detection, snow discrimination, and gap-filling (including hidden Markov random field and snow-depth interpolation techniques). Their releases and updates promoted the research of snow cover characteristics in China. Qiu et al. (2017) yielded a daily FSC dataset with detailed snow cover information over High Mountain Asia with MDC and spatial filtering. Additionally, the global cloud-gap-filled MODIS NDSI dataset (MOD10A1F) is available online since 2020, where cloud-covered grids in the MODIS Terra NDSI product are filled by retaining clear-sky observations from previous days (Hall and Riggs, 2020). However, this dataset performs poorly in China, where periodic and transient snow is dominant. In general, cloud-free SCA datasets produced by composite algorithms are frequently released, while high-quality cloud-free NDSI datasets are still scarce.

To this end, this study generates a spatiotemporally continuous Terra–Aqua MODIS NDSI product with satisfactory accuracy for China, fully considering the spatio-temporal characteristics of regional snow cover variability. A Spatio-Temporal Adaptive fusion method with erroR correction (STAR) improved from our previous work (Jing et al., 2019) is utilized to eliminate cloud obscuration. The long-term detailed snow cover extent dataset will facilitate snow-related scientific studies and practical applications in China.

The rest of this paper is arranged as follows. Firstly, Section 2 describes the input data and the proposed cloud removal method. Then, Section 3 presents the verification accuracy of STAR NDSI collection, with a subsequent analytical application. The cloud removal effectiveness under different cloud coverages is discussed in Section 4. Finally, the data availability and the conclusions are provided in Section 5 and Section 6, respectively.

**Table 1. Typical long-term cloud-free snow cover products covering most snow-dominated regions in China.**

| References | Type | Spatial coverage | Temporal coverage | Temporal resolution | Spatial resolution | DOI |
|---|---|---|---|---|---|---|
| Hao et al. (2021) | SCA | China | 1981–2019 | Daily | ~5 km | 10.11888/Snow.tpdc.271381 |
| Hao et al. (2022) | SCA | China | 2000–2020 | Daily | ~500 m | 10.12072/ncdc.I-SNOW.db0001.2020 |
| Huang (2020) | SCA | Northern hemisphere | 2000–2015 | Daily | ~1 km | 10.12072/ncdc.CCI.db0044.2020 |
| Muhammad and Thapa (2021) | SCA | High Mountain Asia | 2002–2019 | Daily | ~500 m | 10.1594/PANGAEA.918198 |
| Qiu et al. (2017)* | FSC | High Mountain Asia | 2002–2018 | Daily | ~500 m | 10.11922/sciencedb.457 |
| Hall and Riggs (2020) | NDSI | Global coverage | 2000–present | Daily | ~500 m | 10.5067/MODIS/MOD10A1F.061 |

*Cloud coverage is less than 10%.

## 2 Data and Methods

### 2.1 Data

MODIS sensors onboard Terra and Aqua satellites provide the global snow cover datasets. The daily MOD10A1 and MYD10A1 datasets of C6 are available through the website of the National Aeronautics and Space Administration (NASA, https://search.earthdata.nasa.gov/). The NDSI_Snow_Cover (hereafter referred to as NDSI) scientific data set with a range of 0, 10 to 100 was used in this study. As shown in Fig. 1, the NDSI of 19 tiles covering China (excluding sea area) from 1 August 2000 to 31 July 2020, was acquired to generate snow cover maps. The 90 m digital elevation model (DEM) dataset of Shuttle Radar Topographic Mission (SRTM) was obtained from the United States Geological Survey (USGS). In addition, two existing cloud-free snow cover products, including the daily snow cover extent at a 5 km resolution (NIEER AVHRR SCE, https://doi.org/10.11888/Snow.tpdc.271381, Hao et al., 2021) and the daily cloud-gap-filled MODIS NDSI at a 500 m resolution (MODIS CGF NDSI/MOD10A1F, https://doi.org/10.5067/MODIS/MOD10A1F.061, Hall and Riggs, 2020), were used for comparison. For reference data, the snow depth measurements respectively derived from 49 and 102 meteorological stations in NX and QTP (Tibet Meteorological Bureau and National Meteorological Information, 2018) were used for station-based validation. Since the snow depth data can only assess the classification performance of MODIS NDSI retrievals, the NDSI maps derived from Landsat OLI images were utilized for comprehensive validation.

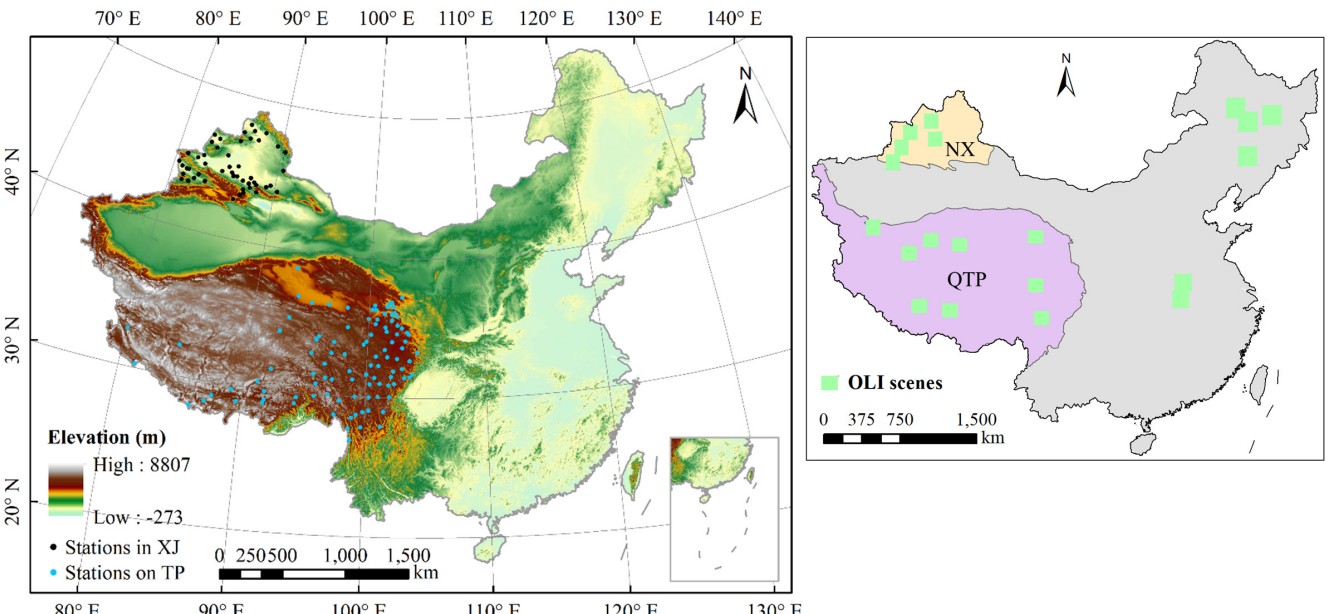

Figure 1. Topographic relief of China, meteorological stations in NX and QTP, and Landsat OLI scenes used for validation.

### 2.2 Algorithm description

MODIS NDSI datasets are unable to represent the daily conditions of snow accumulation and ablation accurately because the optical remote-sensed images are subject to severe cloud pollution. Therefore, a Spatio-Temporal Adaptive fusion method with erroR correction (STAR), which is derived from our two-stage spatio-temporal fusion method (Jing et al., 2019), is presented

to produce a spatio-temporal continuous snow collection. As shown in Fig. 2, the generation procedure comprises the pre-process TAC and the key-process STAR. Then, a quality assessment (QA) approach is presented to provide a data reliability profile for users. On this basis, post-processing is used to further improve the data quality in individual abnormal areas.

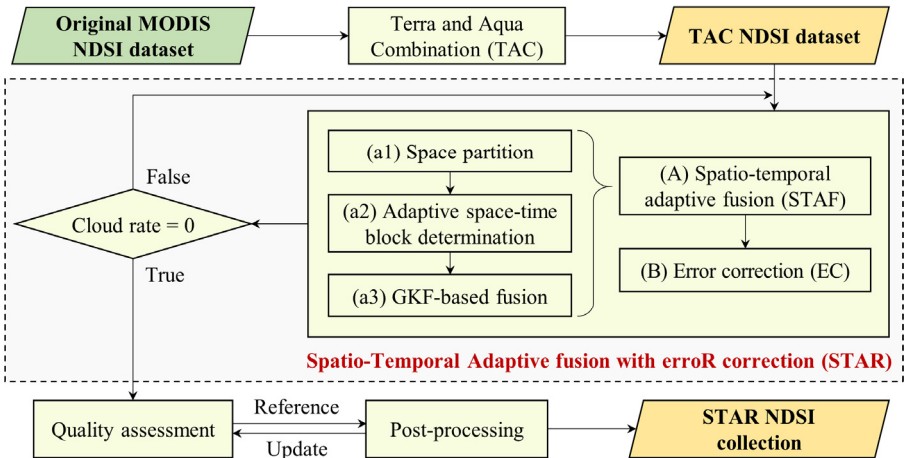

**Figure 2. Schematic of the generation procedure of STAR NDSI collection.**

### 2.2.1 Terra and Aqua combination (TAC)

TAC blends the same-day snow maps deriving from MODIS sensors onboard Terra and Aqua satellites. Its cornerstone is the unlikely significant changes of the snow pattern within the data-acquired time interval (approximately 3 h). Since TAC can efficiently decrease the cloud fraction by 5%–20% with negligible precision sacrifice (Li et al., 2019), it is introduced as a pre-150 processing to reduce cloud coverage preliminarily. Its priority scheme is determined as high value > low value > cloud.

$$NDSI^{P} = NDSI^{Terra} \text{ IF } \left( NDSI^{Terra} \geq NDSI^{Aqua} \text{ OR } NDSI^{Aqua} \text{ is cloud} \right),$$
$$NDSI^{P} = NDSI^{Aqua} \text{ IF } \left( NDSI^{Aqua} > NDSI^{Terra} \text{ OR } NDSI^{Terra} \text{ is cloud} \right),$$

(1)

where $NDSI^{Terra}$ and $NDSI^{Aqua}$ are MODIS NDSI datasets from Terra and Aqua satellites, respectively. $NDSI^{P}$ represents the pre-processed NDSI maps after TAC (referred to as TAC NDSI dataset in subsequent sections). The snow in low altitude and low latitude areas during summer is reversed to no snow to alleviate commission errors inherited from the original data.

In addition, since the Aqua dataset is available since July 2002, the key-process STAR is directly used to remove clouds from Terra MODIS NDSI dataset between August 2000 and May 2002. Particularly, the improved Aqua MODIS C6 NDSI dataset significantly enhances the effectiveness of TAC due to the successful restoration of the absent Aqua MODIS band 6 data by the quantitative image restoration method (Gladkova et al., 2012).

### 2.2.2 Spatio-Temporal Adaptive fusion with erroR correction (STAR)

Many regions with persistent clouds are out of the scope of TAC. To this end, an advanced STAR method, which comprehensively utilizes spatio-temporal contextual information, is proposed to remove the clouds thoroughly. As shown in Fig. 3, the method performs in two passes: spatio-temporal adaptive fusion (STAF) and error correction (EC).

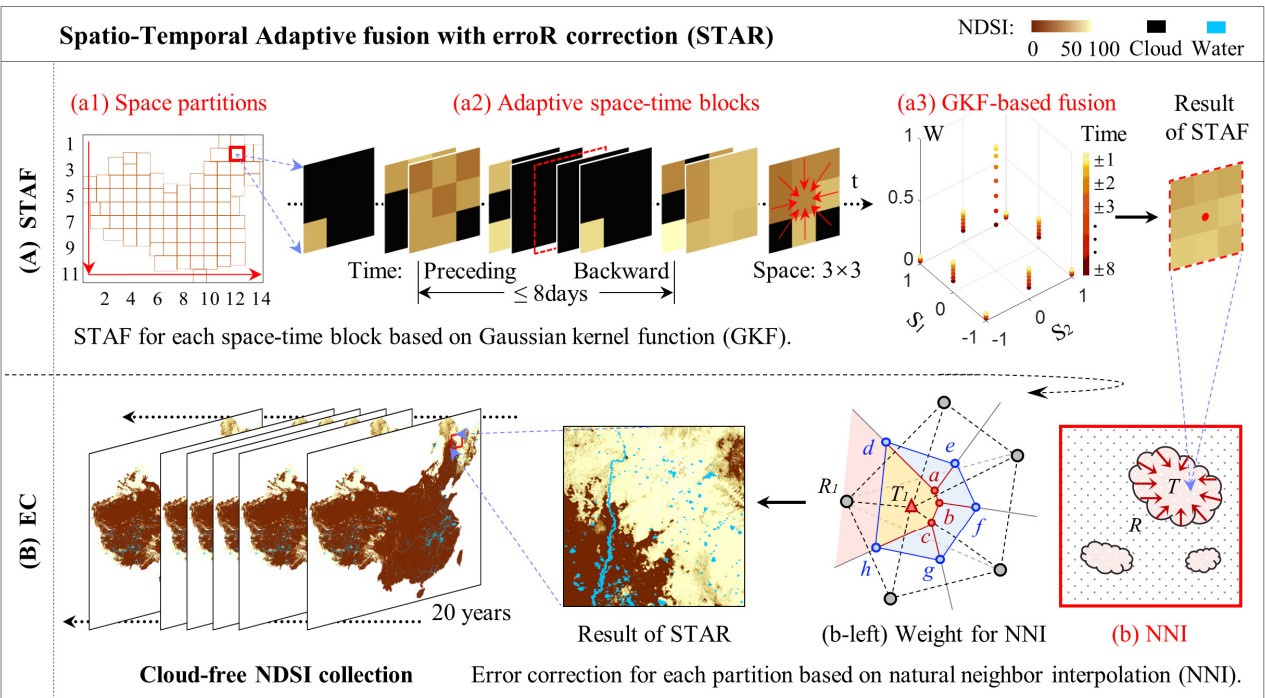

Figure 3. Detailed flowchart of the Spatio-Temporal Adaptive fusion with erroR correction (STAR).

The first pass involves the generation of new NDSI maps by adaptively merging the spatio-temporal contextual information, including space partition, adaptive space-time block determination, and Gaussian kernel function (GKF)-based fusion. The research area is first segmented into dozens of partitions considering the spatial heterogeneity of snow patterns. Thus, the subsequent processes can be performed on a partition basis. Moreover, the optimal query partitions ($Q$) to each target partition ($T$) are determined by a comprehensive consideration of temporal distance ($t$), regional correlation ($r$), and cloud-free fraction ($f$) concerning the temporal complexity of snow variations. The following optimal parameters are derived from the extensive experiments.

$$\begin{cases} Scheme\ 1: r > 0.7,\ if\ f^{C\&T} > 0.3 \\ Scheme\ 2: \max(t^{-1} + f^{C}),\ others \end{cases}, \tag{2}$$

where the regional correlation between the candidate partition ($C$) within an eight-day window and the target partition is considered representative if the fraction of the intersecting cloud-free areas ($f^{C\&T}$) is higher than 0.3. The candidate partition is then determined as a query partition according to Scheme 1 when the regional correlation is larger than 0.7. Otherwise, Scheme 2 is activated. Two query partitions with short distance and high cloud-free fraction are identified within the preceding eight days and the backward eight days, respectively. Subsequently, the 3 × 3 neighborhoods for each pixel of the target partition in all the associated query partitions are determined as the space-time reference block. Last, each pixel is reassigned a fused value from the related space-time block, as expressed in Eq. (3):

$$NDSI_i^F = \sum_{t=1}^{M} \sum_{s=1}^{N} w_{(i,st)} \times NDSI_{(i,st)}^Q,$$

$$where\ W_{(i,st)} = r_t^2 \times \exp\left(\frac{-((\varepsilon \times \Delta s_{(i,s)})^2 + \Delta t_{(i,t)}^2))}{2 \times \sigma^2}\right), \tag{3}$$

where $NDSI_i^F$ denotes the fused NDSI of Pixel $i$ in the target partition. $NDSI_{(i,st)}^Q$ is the pre-processed NDSI in associated query partitions. $M$ is the number of query partitions, each of which contains $N$ reference pixels. In addition, the weight $W_{(i,st)}$ is assigned by a two-dimensional GKF involving the spatial distance ($\Delta s_{(i,s)}$) and the temporal distance ($\Delta t_{(i,t)}$), which is then normalized to $w_{(i,st)}$. $\sigma$ is the standard deviation of GKF. $\varepsilon$ characterizes the dimensional difference, which is equal to $\sigma_t/\sigma_s$ with an expression of each single-dimensional GKF. $r_t$ represents the regional correlation between the query and target partitions if Scheme 1 works; otherwise, it is ignored (i.e., $r_t = 1$). The constant term ($\varepsilon/(2\pi\sigma^2)$) of GKF is ignored due to the normalization process. The important parameters in STAF are listed in Table 2.

**Table 2. Description and default values of STAF parameters.**

| Parameter | Description | Value |
|---|---|---|
| $W_T$ | Temporal window for query partition | $\pm 8$ day |
| $W_N$ | Neighboring window for reference pixel | $3 \times 3$ |
| $r$ | Minimum regional correlation for query partition | 0.7 |
| $\sigma$ | Standard deviation in the GKF | 0.5 |
| $\varepsilon$ | Dimensional difference coefficient ($\sigma_t/\sigma_s$ )in the GKF | 25/9 |

The second pass corrects the fused NDSI maps considering the spatial correlation within a partition. Specifically, the residual errors of the intersecting cloud-free areas of the pre-processed and fused NDSI maps ($NDSI^P$ after TAC and $NDSI^F$ after STAF) are diffused to other cloud-free areas of the fused NDSI maps using the triangulation-based natural neighbor interpolation (Sibson, 1981). Then, the high-quality NDSI maps ($NDSI^H$) can be generated by removing all errors from the fused NDSI maps. The process is formulated as follows:

$$\begin{cases} E_R = NDSI_R^F - NDSI_R^P \\ E_{T(i)} = \sum_{n=1}^{N'} \phi_{(i,n)} E_{R(i,n)}, \\ NDSI_T^H = NDSI_T^F - E_T \end{cases} \tag{4}$$

where $R$ indicates the reference area which is the boundary of the intersecting cloud-free areas. $T$ indicates the target area. The dynamic weights in the error diffusion from $E_R$ to $E_T$ are based on the Voronoi diagrams. As expressed in Fig. 3 (b-left), the original Voronoi cells (bounded by red and gray solid lines) of the reference pixels (gray dots) intersect with the new Voronoi cells (bounded by blue and gray solid lines) of the reference and target pixels. Taking the target pixel $T_1$ with the reference pixel $R_1$ as an example, the weight is assigned as the ratio of the area of the intersecting Voronoi cell ($A_{dabch}$) to that of the new Voronoi cell ($A_{defgh}$).

$$\phi_{(1,1)} = \frac{A_{dabch}}{A_{defgh}}. \tag{5}$$

After all the partitions are processed in sequence, the next iteration of STAR begins until the clouds are completely removed.

### 2.2.3 Quality assessment (QA) approach

A revised QA approach for the gap-filled NDSI collection is proposed on the basis of the quality estimate of MODIS NDSI datasets (Riggs and Hall, 2015), and an example is presented in Fig. A1 (Appendix A). Users can examine the basic quality of the gap-filled NDSI collection considering cloud coverage and spatio-temporal consistency of the raw NDSI dataset by retrieving the bit flags from the integer stored in QA maps. The specific attributes are listed in Table 3.

**Table 3. Bit flags indicating the retrieval conditions according to the raw NDSI dataset.**

| Bit | Description | Bit Combination | Quality |
|-----|-------------|-----------------|---------|
| 0–1 | Comprehensive estimate | 00 | Best |
|     |             | 01 | Good |
|     |             | 10 | OK |
|     |             | 11 | Poor |
| 2   | Pre-processing | 0 | None |
|     |             | 1 | Snow detection reversed |
| 3   | Post-processing | 0 | None |
|     |             | 1 | Post-processed |
| 4   | Number of iterations | 0 | No more than 3 |
|     |             | 1 | More than 3 |
| 5   | Consistency between reference values | 0 | Consistent |
|     |             | 1 | Inconsistent |
| 6   | Consistency between pre-processed and fused values | 0 | Consistent |
|     |             | 1 | Inconsistent |
| 7   | Cloud coverage of the space-time block | 0 | Low [0,60%) |
|     |             | 1 | High [60%,1] |

The snow detection reversal of the pre-processed value in TAC is tracked in Bit 2, the post-processing (Sect. 2.2.4) is tracked in Bit 3, and the number of iterations primarily related to cloud coverage is indicated by Bit 4. If the range of reference values is larger than 30, then Bit 5 flag is set; if the difference between pre-processed and fused values is larger than 20, then Bit 6 flag is set. Bit 7 reflects the cloud coverage of the space-time block. Furthermore, the combination of Bits 0 and 1 is a qualitative estimate of the cloud-removed NDSI collection based on the number of iterations (from here on referred to as NI) and spatio-temporal consistency. The comprehensive estimate is determined as follows:

— if NI = 5 and Bit 5 or 6 is set to 1, then it is assigned "poor";

— if NI = 4 and Bit 5 or 6 is set to 1, then it is assigned "OK";

— if NI = 3 and Bit 5 or 6 is set to 1, then it is assigned "good";

— otherwise, it is assigned "best".

The QA maps are recommended for in-depth application of the cloud-removed NDSI collection.

## 2.2.4 Post-processing

For areas with extremely rapid and fluctuating snow variation, the temporal contextual references are likely to introduce incorrect information and magnify errors in the iterative process. Post-processing is used in this study to reduce the "disorder" phenomenon referring to QA maps. Firstly, the NDSI map with the most consistent snow pattern in adjacent time is artificially identified as a reference. Subsequently, the aforementioned EC is applied to improve the spatial consistency between the post-processing and original areas. Finally, the QA maps are updated.

## 2.3 Validation of the NDSI collection

The gap-filled NDSI collection is evaluated with the in-situ snow depth observations and Landsat NDSI maps considering classification and numerical accuracies according to the current mature verification methods. As shown in Table 4, the classification metrics based on the confusion matrix include overall accuracy (OA), commission error (CE), and omission error (OE) (Klein and Barnett, 2003). Moreover, three general metrics are introduced to measure numerical accuracy: correlation coefficient (CC), absolute error (AE), and root-mean-square error (RMSE).

**Table 4. Confusion matrix and validation metrics.**

| In-situ observations | MODIS NDSI datasets | |
|---|---|---|
| | NDSI > 0 | NDSI = 0 |
| Snow depth > 0 cm | SS | SN |
| Snow depth = 0 cm | NS | NN |

$$OA = \frac{SS + NN}{SS + SN + NS + NN} \qquad CE = \frac{NS}{NS + NN} \qquad OE = \frac{SN}{SN + SS}$$

## 3 Results

As mentioned above, the generation procedure of continuous snow collection includes the pre-process TAC and the key-process STAR. The remainder clouds of 30.62% in the entire collection after TAC are completely removed by STAR. To elaborate the reliability of STAR NDSI collection, TAC NDSI, NIEER AVHRR SCE, and MODIS CGF NDSI products are used as baseline data. Specifically, based on in-situ snow depth measurements, the classification accuracy of STAR NDSI collection is compared with those of TAC NDSI and NIEER AVHRR SCE datasets. In addition, based on Landsat NDSI maps, its numerical accuracy is compared with those of TAC NDSI and MODIS CGF NDSI datasets. This section presents the evaluation results, followed by an exemplary application.

## 3.1 Validation against in-situ snow depth measurements

As described above, the in-situ snow depth data in NX from 1 January 2001 to 31 August 2007 and on QTP from 1 August 2000 to 31 December 2013, were used as the ground truth to evaluate the classification accuracy of TAC NDSI, NIEER

AVHRR SCE, and STAR NDSI datasets. The nearest pixel was matched with each meteorological station, with a total of about 550000 data pairs. Snow-covered pixels in NDSI datasets range from 10 to 100, whereas snow-free pixels are 0; thus, the classification threshold is set as 10 (Zhang et al., 2019). The discriminant threshold for in-situ snow depth is set as 0 or 1 cm. In addition, the cloud-covered areas in TAC NDSI dataset are considered to be snow-free.

Table 5 demonstrates that NIEER AVHRR SCE and STAR NDSI datasets preeminently capture the snow dynamics in
NX referring to the in-situ measurements, with OAs of more than 94%. However, TAC NDSI dataset is insufficient to accurately describe the snow cover variability. Although CEs perform well regardless of the snow depth threshold, OEs of TAC NDSI collection are extremely high, indicating that many cloud-covered areas are dominated by snow. NIEER AVHRR SCE dataset partially retrieves snow pixel under cloud obstruction with an OE decreased by ~43%. STAR NDSI collection completely removes clouds and accurately presents snow distribution, with an OE further decreased from ~17% to ~7%. The
generation procedure in NX has two strengths. Firstly, the satellite-borne sensors can accurately capture the snow events on the ground due to the generally thick snow averaging approximately 20 cm. Secondly, the gap-filling approach with comprehensive consideration of spatial and temporal correlation has outstanding reliability due to the significant periodicity of snow variation. It can be inferred that the NDSI datasets in NC have high accuracy because of the similar snow conditions, despite the lack of in-situ data in this region.

By contrast, despite the satisfactory performance of OAs and CEs, the OEs of three snow cover datasets over QTP are as remarkably high as 72%, 42%, and 39% even at the snow depth threshold of 1 cm (Table 6). This finding indicates the omission of a large number of snow-covered pixels. The specific reasons are as follows. Firstly, the original MODIS NDSI maps frequently underestimate the snow presence throughout the snow period because discriminating the shallow snow pixels with an averaged snow depth of approximately 4 cm over QTP is challenging. Secondly, the credibility of the spatio-temporal
contextual information is relatively low because the patchy snow rapidly and irregularly varies due to the extremely complex topographic and climatic conditions, leading to a further decrease in the accuracy of the gap-filled results. Lastly, the meteorological stations over QTP are unevenly distributed and are mostly located in low- and medium-altitude/latitude areas dominated by transient snow. Consequently, the evaluation results slightly exaggerate the real OEs.

For the in-depth analysis of the temporal characteristics, the monthly classification accuracies of TAC NDSI, NIEER
AVHRR SCE and STAR NDSI products in NX and QTP are shown in Fig. 4 (the horizontal axis is the month in a hydrological year). In NX (group a), the monthly snow fraction in the in-situ samples is greater than 85% from December to next February. Therefore, the clouds in TAC NDSI dataset seriously affect the snow cover estimation, while both cloud-free products exhibit superior OAs. Compared to NIEER AVHRR SCE product, STAR NDSI collection has slightly higher CEs but relatively lower OEs. The OEs of STAR NDSI collection typically occur during snow accumulation and ablation periods, and almost disappear
during stable snow-cover and snow-free periods. In QTP, the snow period is generally from October to next May, with the monthly snow fraction of less than 10% in the in-situ samples. Consequently, the underestimation of snow coverage caused

by the clouds in TAC NDSI dataset is slight. All three products achieve outstanding OAs and CEs, but exhibit relatively poor

OEs. In NIEER AVHRR SCE and STAR NDSI datasets, these OEs are generally observed outside the snow period. As

mentioned above, there are three reasons for this phenomenon. Nonetheless, STAR NDSI collection presents superior

classification accuracy to TAC NDSI and NIEER AVHRR SCE datasets.

**Table 5. Confusion matrices between TAC NDSI, NIEER AVHRR SCE, STAR NDSI datasets and in-situ snow-depth (SD) data in NX from 1 January 2001 to 31 August 2007.**

| Station | TAC | | | NIEER | | | STAR | | |
|---|---|---|---|---|---|---|---|---|---|
| **Indicator** | Snow | No snow | Total | Snow | No snow | Total | Snow | No snow | Total |
| Snow (SD > 0 cm) | 13466 | 19836 | 33302 | 27656 | 5646 | 33302 | 30955 | 2347 | 33302 |
| **OE** | 40% | **60%** | | 83% | **17%** | | 93% | **7%** | |
| No snow | 1269 | 76909 | 78178 | 1384 | 76794 | 78178 | 2741 | 75437 | 78178 |
| **CE** | **2%** | 98% | | **2%** | 98% | | **4%** | 96% | |
| Total | 14735 | 96745 | 111480 | 29040 | 82440 | 111480 | 33696 | 77784 | 111480 |
| **OA** | | **81%** | | | **94%** | | | **95%** | |
| Snow (SD > 1 cm) | 13136 | 18390 | 31526 | 26899 | 4627 | 31526 | 29909 | 1617 | 31526 |
| **OE** | 42% | **58%** | | 85% | **15%** | | 95% | **5%** | |
| No snow | 1599 | 78355 | 79954 | 2141 | 77813 | 79954 | 3787 | 76167 | 79954 |
| **CE** | **2%** | 98% | | **3%** | 97% | | **5%** | 95% | |
| Total | 14735 | 96745 | 111480 | 29040 | 82440 | 111480 | 33696 | 77784 | 111480 |
| **OA** | | **82%** | | | **94%** | | | **95%** | |

**Table 6. Confusion matrices between TAC NDSI, NIEER AVHRR SCE, STAR NDSI datasets and in-situ snow-depth (SD) data in QTP from 1 August 2000 to 31 December 2013.**

| Station | TAC | | | NIEER | | | STAR | | |
|---|---|---|---|---|---|---|---|---|---|
| **Indicator** | Snow | No snow | Total | Snow | No snow | Total | Snow | No snow | Total |
| Snow (SD > 0 cm) | 5189 | 18095 | 23284 | 10900 | 12384 | 23284 | 11219 | 12065 | 23284 |
| **OE** | 22% | **78%** | | 47% | **53%** | | 48% | **52%** | |
| No snow | 6145 | 404624 | 410769 | 17663 | 393106 | 410769 | 9338 | 401431 | 410769 |
| **CE** | **1%** | 99% | | **4%** | 96% | | **2%** | 98% | |
| Total | 11334 | 422719 | 434053 | 28563 | 405490 | 434053 | 20557 | 413496 | 434053 |
| **OA** | | **94%** | | | **93%** | | | **95%** | |
| Snow (SD > 1 cm) | 4126 | 10357 | 14483 | 8391 | 6092 | 14483 | 8813 | 5670 | 14483 |
| **OE** | 28% | **72%** | | 58% | **42%** | | 61% | **39%** | |
| No snow | 7208 | 412362 | 419570 | 20172 | 399398 | 419570 | 11744 | 407826 | 419570 |
| **CE** | **2%** | 98% | | **5%** | 95% | | **3%** | 97% | |
| Total | 11334 | 422719 | 434053 | 28563 | 405490 | 434053 | 20557 | 413496 | 434053 |
| **OA** | | **96%** | | | **94%** | | | **96%** | |

Overall, STAR NDSI collection is capable of snow status estimation, eliminating cloud contamination in TAC NDSI

dataset, and capturing more snow events than NIEER AVHRR SCE dataset. However, the accuracy of STAR NDSI collection

has regional and temporal heterogeneity. Firstly, the accuracy over QTP is lower than that of NX, which is consistent with the

characteristic of the original MODIS NDSI maps. Then, the permanent and periodic snow regime regions reconstructed by

STAR have prominently high accuracy, while the transient snow-covered regions are easily omitted. Fortunately, the

monitoring of permanent and periodic snow plays a key role in most snow-related investigations. Finally, the accuracy of stable

snow-cover and snow-free periods is slightly better than that of snow accumulation and ablation periods.

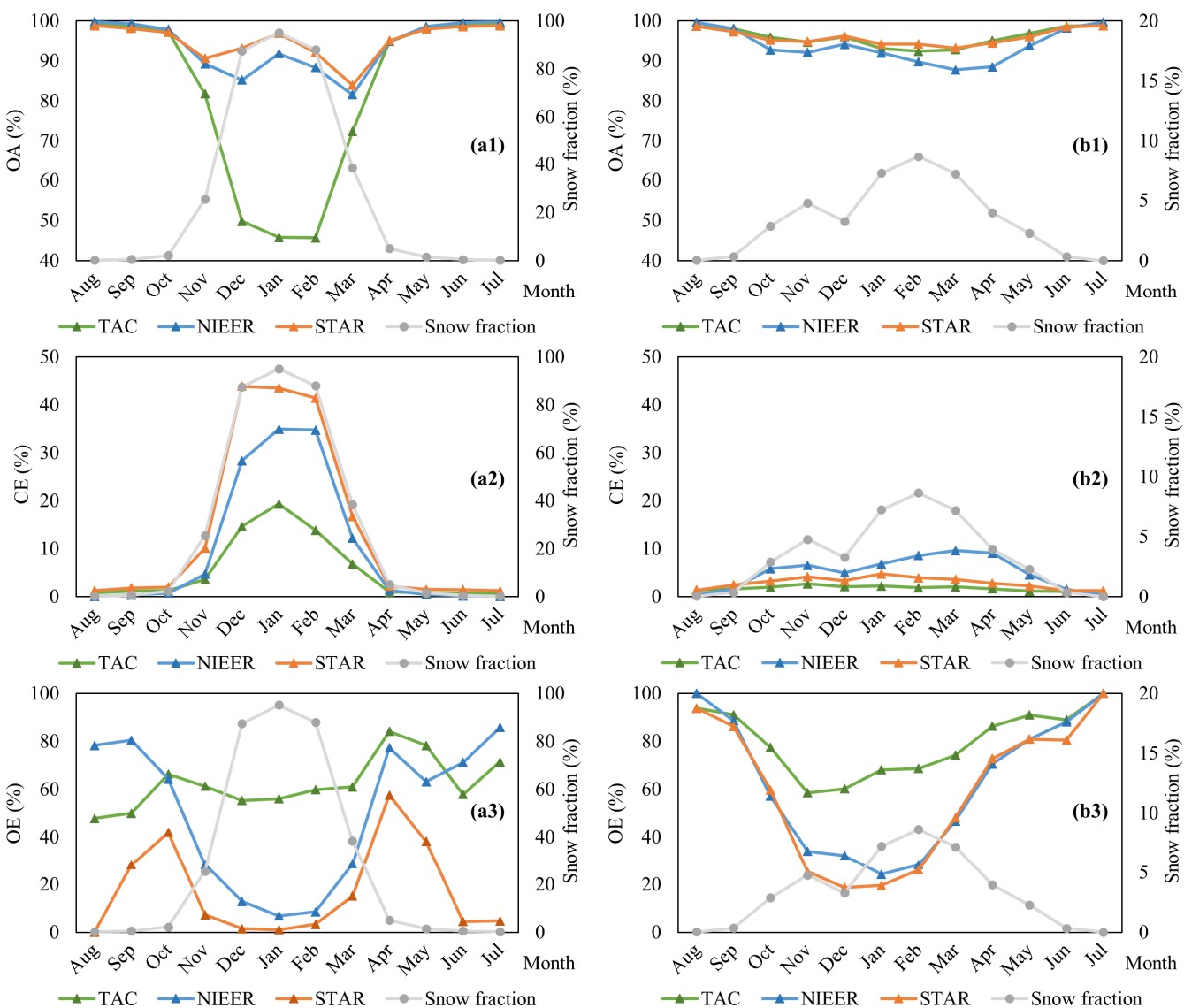

**Figure 4. Monthly classification accuracy of TAC NDSI, NIEER AVHRR SCE, and STAR NDSI products on NX (group a) and QTP (group b). Note that the optimal values for OA, CE and OE are 100%, 0% and 0%, respectively.**

## 3.2 Validation based on Landsat NDSI maps

Only the classification accuracy can be evaluated by in-situ measurements due to the significant difference in the nature of the

snow depth and NDSI data. Therefore, Landsat images with finer spatial resolution were commonly adopted for the numerical

evaluation of NDSI datasets (Crawford, 2015). NDSI values for the Landsat 8 dataset were calculated as follows:

$\left(Band3-Band6\right)\big/\left(Band3+Band6\right)$. Subsequently, the average of the 17 × 17 neighborhoods closest to the center of the

MODIS NDSI pixel in the Landsat NDSI map was considered to be the reference of this MODIS NDSI pixel to match the

spatial resolution of Landsat with that of MODIS. Specifically, the cloud-contaminated pixels marked by the quality band in

Landsat images were excluded, and the reference areas with cloud coverage larger than 30% were discarded. A total of 19

Landsat NDSI maps with different snow coverages from January to April in 2018, which are distributed in NC (4 scenes),

Central China region (CCR, 2 scenes), QTP (8 scenes), and NX (5 scenes), were exploited as evaluation references for this

validation experiment. Two evaluations including a cross-comparison of TAC NDSI, MODIS CGF NDSI, and STAR NDSI datasets and an internal comparison of clear-sky and cloud-cover areas are described in detail below.

For the cross-comparison, the visual effects of three NDSI datasets on 8 January 2018 and 3 February 2018 are shown in Fig. 5. TAC NDSI dataset is still heavily obscured by clouds. Although MODIS CGF NDSI dataset completely removes clouds from the MOD10A1 product, it is difficult to accurately retrieve periodic and transient snow cover areas due to the simplicity of the cloud-gap-filled method. Specifically, the gaps are filled by retaining clear-sky observations from previous days. However, snow patterns under cloud cover are likely to change significantly during these days. Therefore, snow cover is significantly underestimated during accumulation (Fig. 5, a2) and overestimated during ablation (Fig. 5, b2). By contrast, STAR NDSI collection preeminently captures the snow dynamics under temporally continuous clouds, attributing to the spatio-temporal adaptive fusion strategy. Furthermore, the three NDSI datasets are quantitatively assessed by Landsat NDSI maps.

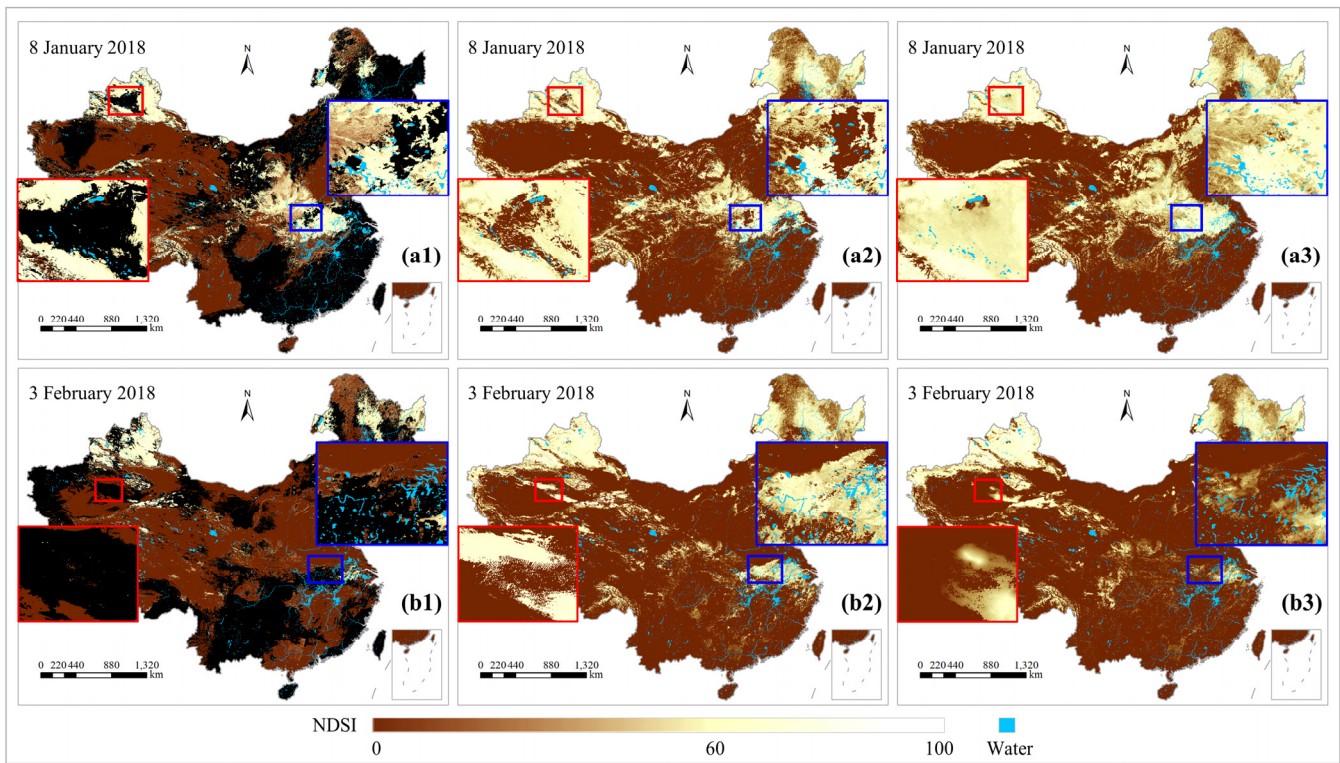

**Figure 5. Comparison of TAC NDSI (column 1), MODIS CGF NDSI (column 2), and STAR NDSI (column 3) products on 8 January 2018 (group a) and 3 February 2018 (group b).**

As the average cloud cover is as high as 40.7%, TAC NDSI dataset has a low correlation with Landsat NDSI maps, with an average CC of 0.46 (Table 7). The cloud-free MODIS CGF NDSI dataset enhances the correlation with Landsat NDSI maps, with an average CC of 0.73. By contrast, the snow dynamics presented by the spatiotemporally continuous STAR NDSI dataset are highly consistent with Landsat NDSI maps, with an average CC further increased to 0.84. Compared with TAC NDSI and MODIS CGF NDSI datasets, the average RMSE of STAR NDSI dataset is decreased by 9.06 and 5.58, respectively. The positive AEs reveal that NDSI values for snow pixels in MODIS CGF NDSI and STAR NDSI datasets are generally higher

than those of Landsat NDSI maps. In terms of snow coverage, STAR NDSI dataset notably improves the detection of snow

events compared to the other two datasets, with an average absolute SRD decreased by 31.1% and 2.4% (SRD indicates the

difference of snow rate between MODIS and Landsat NDSI datasets). Consequently, STAR NDSI collection is a more

promising snow cover product than TAC NDSI and MODIS CGF NDSI datasets, contributing to related hydrological and

meteorological studies.

**Table 7. Performance statistics for TAC NDSI, MODIS CGF NDSI, and STAR NDSI datasets against Landsat NDSI maps.**

| Region_Date | Cloud cover (%) | CC | | | RMSE | | | AE | | | SRD (%) | | |
|---|---|---|---|---|---|---|---|---|---|---|---|---|---|
| | | TAC | CGF | STAR | TAC | CGF | STAR | TAC | CGF | STAR | TAC | CGF | STAR |
| NC1_20180225 | 61.4 | 0.49 | 0.90 | 0.87 | 25.64 | 20.20 | 17.10 | -11.59 | 18.35 | 15.07 | -60.1 | 1.0 | 1.0 |
| NC2_20180311 | 43.6 | 0.69 | 0.72 | 0.83 | 26.75 | 19.77 | 13.87 | -9.36 | 15.72 | 12.14 | -43.1 | -5.7 | -1.7 |
| NC3_20180311 | 34.6 | 0.19 | 0.86 | 0.86 | 18.61 | 20.97 | 8.79 | -10.26 | 10.78 | 0.15 | -34.2 | 0.0 | -2.8 |
| NC4_20180318 | 16.3 | 0.73 | 0.95 | 0.98 | 21.50 | 15.03 | 10.25 | -3.33 | 8.26 | 5.42 | -16.6 | 0.5 | -1.1 |
| CCR1_20180203 | 14.9 | 0.20 | 0.93 | 0.95 | 11.53 | 14.91 | 5.04 | -3.56 | 6.55 | 1.54 | -11.4 | 4.2 | 2.8 |
| CCR2_20180203 | 95.0 | -0.07 | 0.52 | 0.73 | 14.26 | 38.77 | 8.43 | -9.07 | 32.76 | 0.10 | -47.8 | 29.8 | -5.4 |
| QTP1_20180322 | 36.3 | 0.39 | 0.75 | 0.83 | 18.03 | 14.26 | 10.70 | -5.30 | 2.02 | 0.77 | -13.8 | 1.2 | 1.3 |
| QTP2_20180225 | 22.4 | 0.54 | 0.71 | 0.82 | 25.50 | 17.92 | 15.27 | -9.66 | -2.62 | -0.30 | -27.8 | -11.5 | -9.1 |
| QTP3_20180320 | 15.3 | 0.29 | 0.31 | 0.74 | 11.40 | 11.64 | 7.91 | -2.89 | -2.37 | -1.49 | -6.7 | -5.9 | -3.5 |
| QTP4_20180401 | 29.5 | 0.47 | 0.51 | 0.79 | 30.94 | 29.60 | 16.64 | -16.86 | -16.36 | -3.71 | -31.6 | -29.2 | -8.3 |
| QTP5_20180307 | 42.5 | 0.47 | 0.92 | 0.92 | 30.00 | 13.80 | 13.80 | -11.76 | 7.67 | 7.67 | -36.0 | 1.0 | 1.0 |
| QTP6_20180305 | 64.9 | 0.17 | 0.76 | 0.78 | 42.26 | 17.26 | 14.53 | -32.30 | -4.49 | 4.67 | -66.1 | -10.9 | -2.9 |
| QTP7_20180107 | 60.8 | 0.44 | 0.75 | 0.80 | 28.13 | 18.09 | 18.08 | -15.95 | -0.48 | 6.04 | -60.0 | -19.2 | -12.6 |
| QTP8_20180128 | 34.6 | 0.49 | 0.38 | 0.82 | 11.98 | 28.32 | 10.65 | 0.67 | -4.91 | 2.68 | 1.1 | -47.7 | 4.3 |
| NX1_20180105 | 52.2 | 0.79 | 0.89 | 0.86 | 27.53 | 22.78 | 22.81 | -4.00 | 20.21 | 22.18 | -52.2 | 8.4 | 0.1 |
| NX2_20180213 | 23.4 | 0.64 | 0.82 | 0.92 | 23.55 | 10.65 | 20.81 | 7.13 | 2.68 | 18.29 | -14.8 | 4.3 | 7.2 |
| NX3_20180220 | 56.0 | 0.56 | 0.73 | 0.86 | 26.35 | 24.83 | 18.78 | -9.87 | 21.24 | 16.06 | -51.2 | 2.8 | 1.9 |
| NX4_20180103 | 23.5 | 0.70 | 0.65 | 0.74 | 28.94 | 25.58 | 28.86 | 15.70 | 22.93 | 26.66 | -21.8 | -1.1 | -0.2 |
| NX5_20180220 | 46.1 | 0.55 | 0.87 | 0.92 | 23.55 | 15.95 | 11.99 | -10.44 | 5.59 | 2.28 | -32.7 | -4.1 | -8.0 |
| Average | 40.7 | **0.46** | **0.73** | **0.84** | **23.50** | **20.02** | **14.44** | **-7.51** | **7.55** | **7.17** | **-33.0** | **-4.3** | **-1.9** |

Note that TAC, CGF, and STAR represent TAC NDSI, MODIS CGF NDSI, and STAR NDSI datasets, respectively.

In addition to the cross-comparison of TAC NDSI, MODIS CGF NDSI, and STAR NDSI datasets, an internal comparison

of STAR NDSI collection in clear-sky areas and cloud-cover areas was performed based on Landsat NDSI maps, to highlight

the accuracy of the recovered pixels in STAR NDSI collection. As described above, clear-sky areas and cloud-cover areas

account for 59.3% and 40.7%, respectively. Table 8 indicates that the snow distribution of recovered areas in STAR NDSI

collection is relatively consistent with that of Landsat NDSI maps. Although the average CC decreases from 0.85 to 0.73 and

the average RMSE increases from 13.48 to 16.30 compared with clear-sky areas, the accuracy of recovered areas is satisfactory.

Since many recovered areas inherit errors from clear-sky areas because the cloud removal procedure completely relies on the

350 original dataset, a slight decrease in accuracy is reasonable. In addition, the average AEs of clear-sky and recovered areas are

7.81 and 6.83, respectively, revealing the systematic overestimation of NDSI values in snow-cover areas (Landsat NDSI values are generally low). Except for a few areas, the snow conditions in most cloud-cover areas are well recovered, with an average SRD of -5.0%. This finding highlights that STAR NDSI collection can completely remove clouds with satisfactory accuracy.

**Table 8. Performance statistics for STAR NDSI collection against Landsat NDSI maps in clear-sky and cloud-cover areas according to the TAC dataset.** Note that red and blue bold values respectively indicate an improvement and degradation of cloud-cover areas compared with clear-sky areas (corresponding to four groups in Fig. 6).

| Region_Date | CC | | RMSE | | AE | | SRD (%) | |
|---|---|---|---|---|---|---|---|---|
| | Clear-sky | Cloud-cover | Clear-sky | Cloud-cover | Clear-sky | Cloud-cover | Clear-sky | Cloud-cover |
| NC1_20180225 | 0.95 | 0.76 | 16.89 | 17.23 | 15.23 | 14.97 | 2.91 | -0.2 |
| NC2_20180311 | 0.89 | 0.71 | 12.27 | 15.71 | 11.78 | 12.62 | -0.10 | -3.7 |
| NC3_20180311 | **0.92** | **0.42** | 2.31 | 14.60 | -0.09 | 0.61 | -1.02 | -6.1 |
| NC4_20180318 | 0.98 | 0.86 | 10.29 | 14.75 | 4.88 | 11.30 | -1.23 | -0.2 |
| CCR1_20180203 | 0.83 | 0.68 | 3.37 | 10.29 | 0.68 | 6.50 | 3.48 | -0.9 |
| CCR2_20180203 | **0.55** | **0.74** | 10.93 | 8.28 | 6.80 | -0.25 | 32.63 | -7.4 |
| QTP1_20180322 | 0.75 | 0.87 | 10.22 | 11.49 | 0.44 | 1.35 | 0.11 | 3.4 |
| QTP2_20180225 | **0.86** | **0.64** | 13.50 | 20.24 | 0.89 | -4.40 | -7.49 | -14.8 |
| QTP3_20180320 | 0.73 | 0.69 | 3.74 | 18.19 | -0.40 | -7.52 | -1.07 | -16.9 |
| QTP4_20180401 | 0.79 | 0.78 | 16.56 | 17.08 | -4.35 | -2.17 | -8.14 | -8.3 |
| QTP5_20180307 | 0.94 | 0.88 | 13.82 | 13.86 | 8.01 | 7.16 | 1.68 | 0.1 |
| QTP6_20180305 | 0.79 | 0.76 | 15.03 | 14.26 | 1.75 | 6.24 | -4.79 | -1.9 |
| QTP7_20180107 | 0.98 | 0.63 | 15.17 | 19.74 | 10.28 | 3.27 | -1.00 | -20.0 |
| QTP8_20180128 | 0.75 | 0.89 | 11.13 | 9.92 | 3.27 | 1.67 | 7.62 | -1.7 |
| NX1_20180105 | 0.89 | 0.62 | 24.47 | 21.17 | 24.29 | 20.25 | 0.00 | 0.1 |
| NX2_20180213 | 0.95 | 0.74 | 20.47 | 21.88 | 18.27 | 18.37 | 9.00 | 1.5 |
| NX3_20180220 | 0.93 | 0.75 | 17.45 | 19.77 | 15.37 | 16.60 | 4.02 | 0.2 |
| NX4_20180103 | **0.64** | **0.58** | 29.54 | 26.53 | 28.09 | 22.00 | 0.61 | -2.9 |
| NX5_20180220 | 0.97 | 0.86 | 8.92 | 14.81 | 3.26 | 1.13 | -1.31 | -15.7 |
| Average | 0.85 | 0.73 | 13.48 | 16.30 | 7.81 | 6.83 | 1.89 | -5.0 |

For in-depth verification analysis, Figure 6 shows the visual effects in four regions corresponding to four highlighted groups in Table 8. The accuracy of clear-sky areas in NC is prominently high with a CC of 0.92, while recovered areas notably reduce the performance with a CC of 0.42. However, Figure 6a shows that clear-sky areas in TAC NDSI dataset cannot reflect the snow events, whereas STAR NDSI collection effectively retrieves these events. Inevitably, the snow edges are slightly inaccurate and blurry due to insufficient reference information. For group (b), the original accuracy of the NDSI dataset in CCR is relatively low, with high cloud coverage and false acceptance rate, while STAR NDSI collection presents a snow pattern consistent with Landsat NDSI. Nevertheless, CCR is a transient snow area with relatively low altitudes and latitudes. Therefore, the gap-filled result has visible uncertainty, in which commission (black box) and omission (red box) frequently occur. As mentioned above, MODIS NDSI datasets generally perform poorly over QTP. However, Figure 6c demonstrates that TAC NDSI and STAR NDSI collections can accurately capture snow events despite a few omissions (red box). Similar to the

NC region, the CCs provide a positive indication of overall performance in NX. As shown in Fig. 6d, even the *NX4_20180103* scene with a remarkably low CC of 0.58 can effectively reflect the snow pattern. In addition, the NDSI dataset retrieved by STAR inevitably has a few extremely abnormal areas during 20 years due to the vast territory of China; an example is presented in Fig. A2 (Appendix A). These areas have severe cloud contamination and irregular snow dynamics, contributing to the challenges in reconstruction and evaluation. Therefore, these areas are corrected by post-processing as described in Sect. 2.2.4.

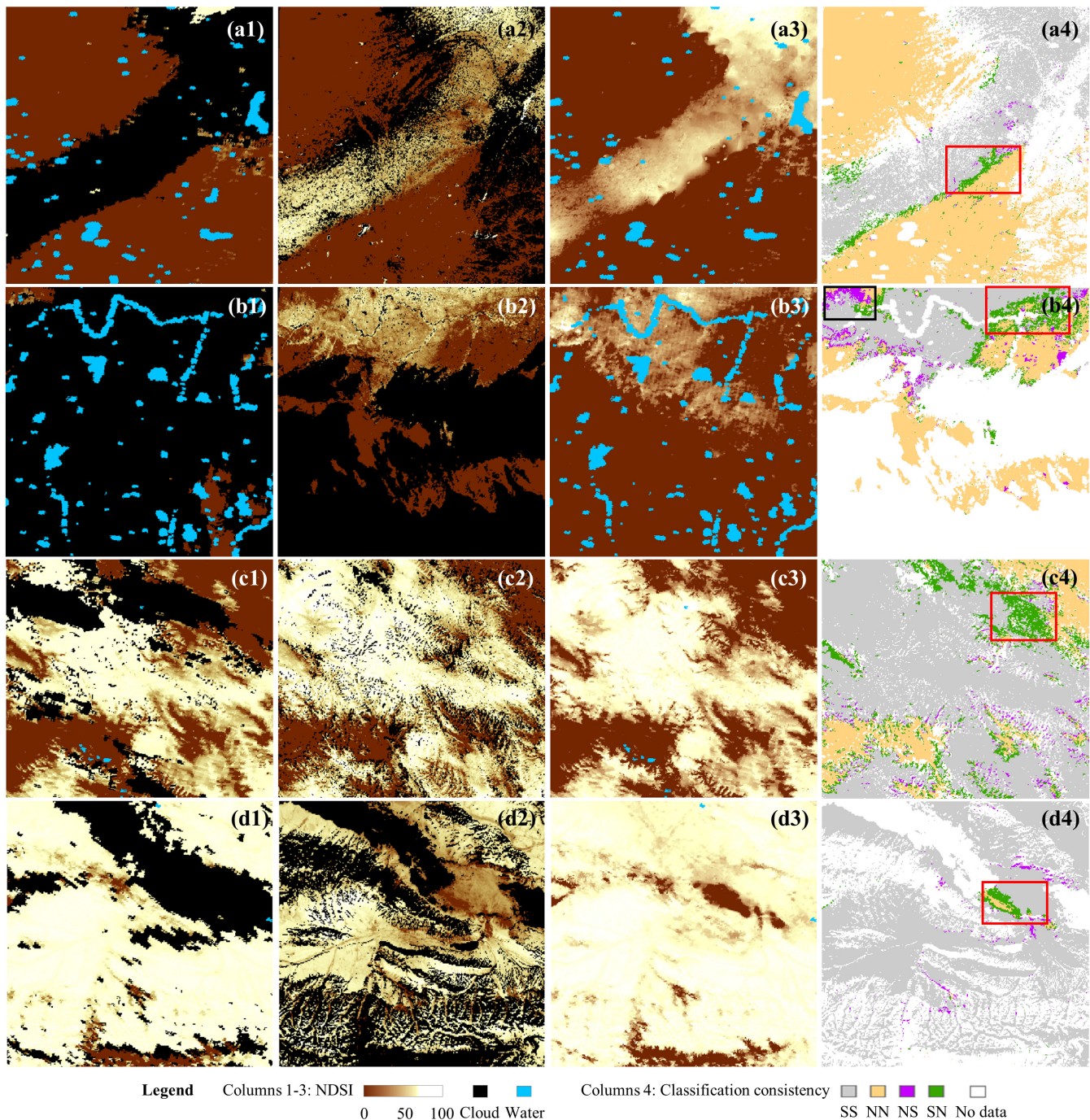

**Figure 6. Comparison of TAC NDSI (column 1), Landsat NDSI (column 2), and STAR NDSI (column 3) products and classification consistency (column 4) corresponding to NC3_20180311, CCR2_20180203, QTP2_20180225, and NX4_20180103 (groups *a* to *d*).**

Overall, the numerical accuracy of STAR NDSI collection is rigorously evaluated based on fine-resolution Landsat NDSI

maps. The cross-comparison indicates that STAR NDSI collection is superior to both TAC NDSI and MODIS CGF NDSI datasets. In addition, the internal comparison reveals that the effectiveness of cloud removal is satisfactory, although recovered areas have slightly lower accuracy than clear-sky areas. Consequently, STAR NDSI collection has considerable application potential.

### 3.3 Application of STAR NDSI collection

In addition to quality evaluation, the exemplary analysis also contributes to understanding the potential of STAR NDSI collection for hydrological and climatic applications. Therefore, the spatial distribution and temporal variability of snow cover in China are analyzed as two simple application demonstrations.

From a spatial perspective, the sequence shown in Fig. 7 indicates that the snow fraction first increases and then decreases in NC and NX regions from 5 December 2014 to 25 January 2015. However, due to the complex topographic and climatic conditions, the snow cover on QTP presents irregular distribution and considerable fluctuations. During this period, the snow cover pattern in China changed dramatically, with the overall snow fraction ranging from 19.25% to 35.42%. In addition, another sequence of cloud-free NDSI maps in early 2020 is shown in Fig. 8. During this period, the snow cover in China presents a single-wave depletion curve. The peaks of snow fractions are observed in three major snow areas between 11 January 2020 and 21 January 2020. Compared with the previous sequence, the snow cover variation on QTP in this sequence has significantly enhanced regularity.

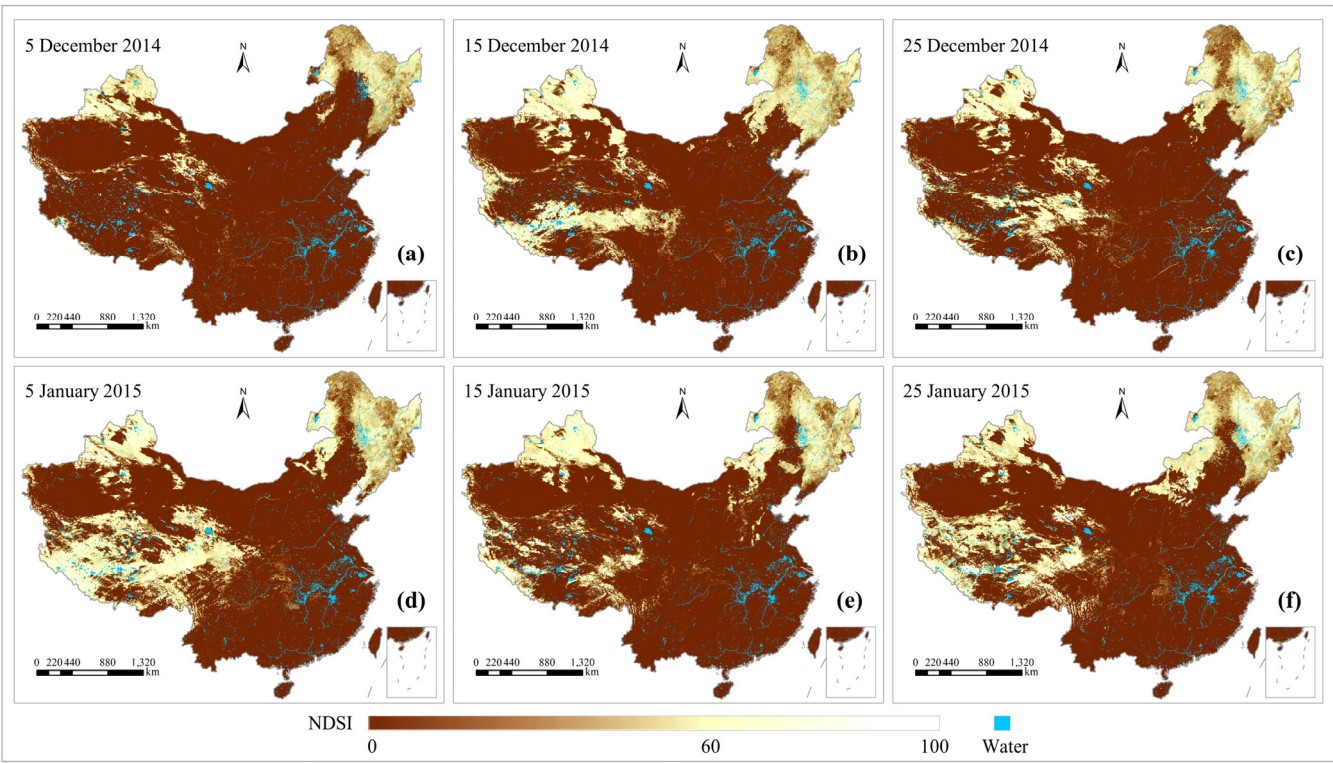

**Figure 7. A sequence of STAR NDSI collection from 5 December 2014 to 25 January 2015.**

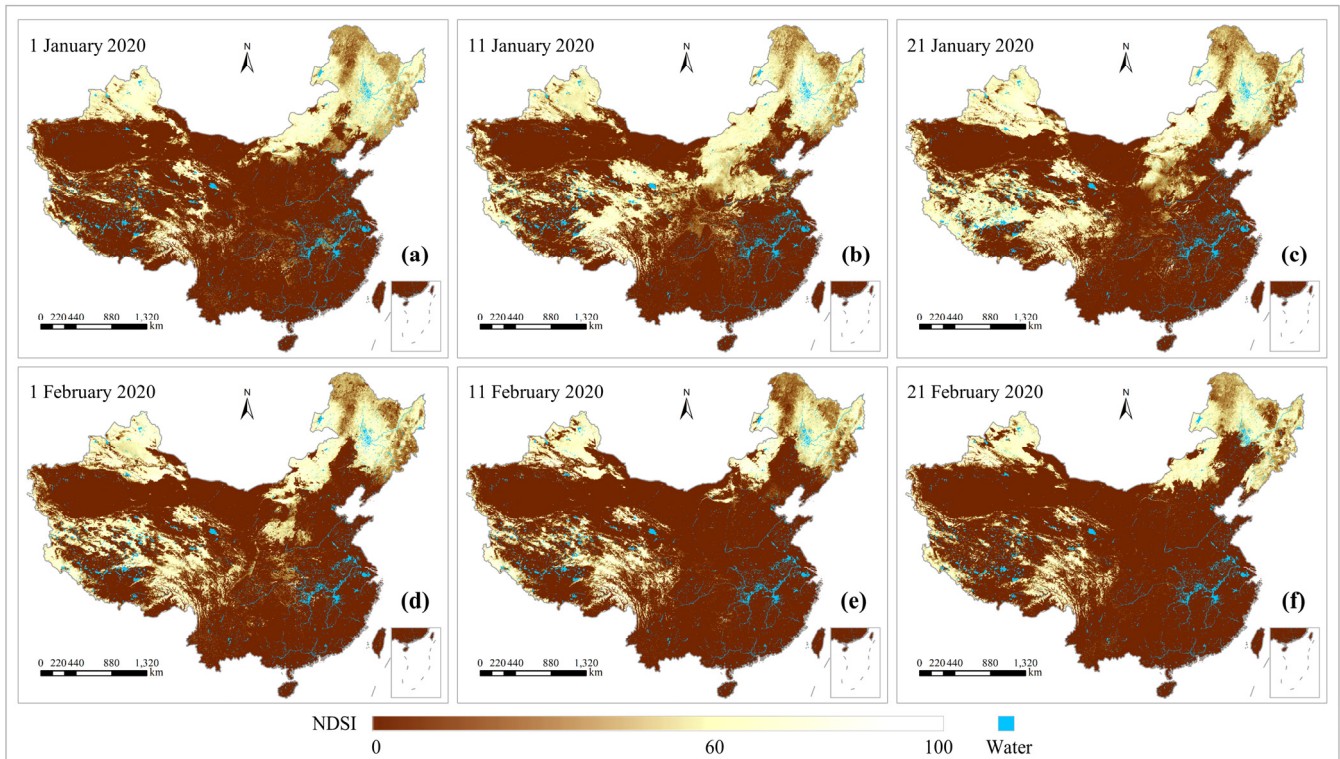

**Figure 8. A sequence of STAR NDSI collection from 1 January 2020 to 21 February 2020.**

Figure 9 shows the daily average snow fraction in the three main subregions in China and the entire situation considering temporal analysis. In terms of intra-annual variability, the snow dynamics periodically evolve in NX and NC but substantially fluctuate on QTP. NX and NC have similar snow depletion curves, demonstrating rapid accumulation and ablation in November and March, respectively. QTP has a relatively long snow period, with an average snow fraction varying from 20% to 40% from October to next May. Consequently, China is dominated by periodic snow. As for inter-annual variability, among the three major snow areas, the snow fraction in NC remarkably fluctuates with a standard deviation of 5.3%. The snow coverage on QTP presented a slight decreasing trend from 2005 to 2017 but increased significantly in the past two years. In particular, rather than a significant rise in maximum snow coverage, the increase can be observed throughout the snow period with a slight reduction in intra-annual volatility. This finding implies that the regional climatic conditions tend to stabilize slightly. In addition, no significant trend has been detected in snow dynamics in China during the 20 years. Nevertheless, the significant fluctuation of maximum snow coverage in China indicates the presence of non-negligible large-scale transient snow cover areas. For example, Figure A3 (Appendix A) shows the extreme snow event in southern China caused by the La Niña phenomenon, which resulted in heavy casualties and economic losses in the hydrological year 2007–2008.

Overall, STAR NDSI collection can accurately reflect the spatial and temporal dynamics of snow cover in China. It is promising for hydrological and meteorological applications.

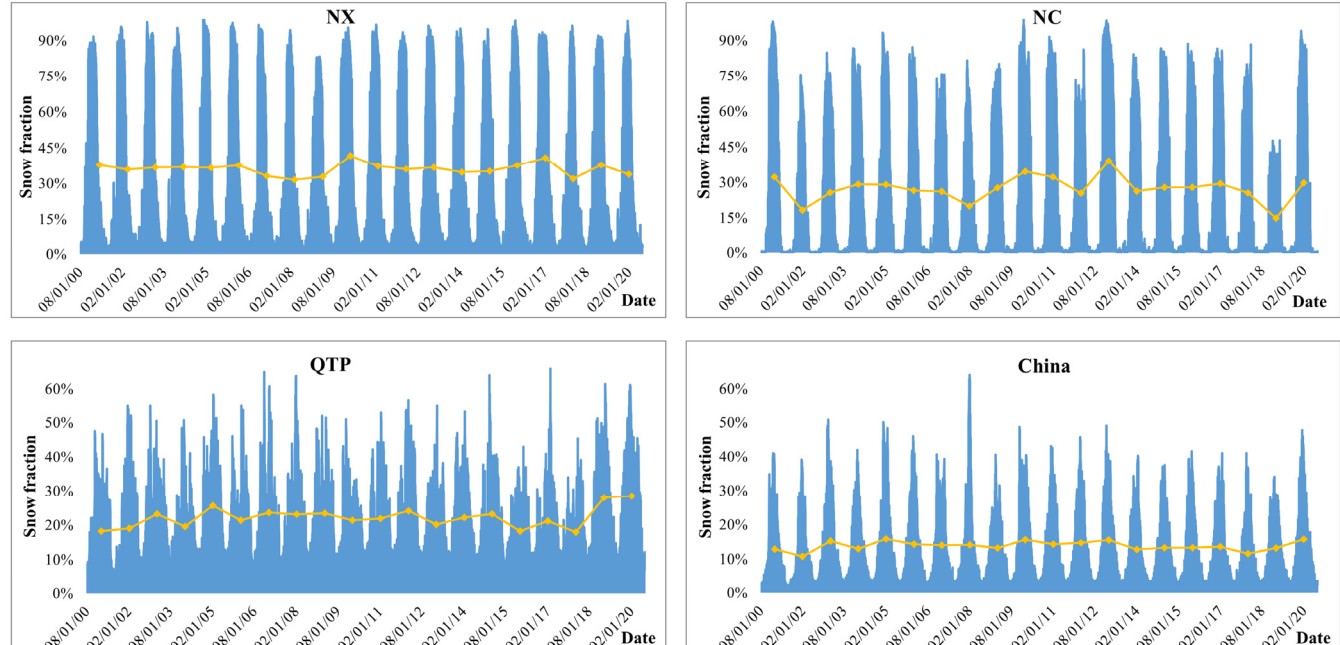

**Figure 9. Daily average snow fraction of NX, NC, QTP, and China.**

## 4 Discussion

To elaborate the cloud removal effectiveness of the proposed STAR method, the performance statistics under different simulated cloud conditions are shown in Table 9. Four TAC NDSI maps with little cloud cover during the snow period were used in the simulated experiment. Cloud masks from other dates were added to the target maps, with different fractions of about 20%, 50% and 80%. Subsequently, the artificially cloud-covered maps were recovered by STAR and validated by Landsat NDSI maps.

**Table 9. The cloud removal effectiveness of STAR compared to Landsat NDSI maps under different simulated cloud conditions.** Note that blue bold values indicate a significant degradation of the accuracy under the current cloud cover compared to the previous one.

| Region_Date | Snow fraction (%) | Added cloud (%) | CC | | RMSE | | AE | | SRD (%) | |
|---|---|---|---|---|---|---|---|---|---|---|
| | | | TAC | STAR | TAC | STAR | TAC | STAR | TAC | STAR |
| NC4_20180318 | 44% | 11% | | 0.98 | | 10.44 | | 4.62 | | -1.78 |
| | | **55%** | 0.98 | **0.89** | 10.29 | **19.11** | 4.88 | 11.17 | -1.23 | **12.96** |
| | | 80% | | 0.86 | | 21.24 | | 13.03 | | 16.31 |
| QTP2_20180225 | 83% | 16% | | 0.85 | | 13.77 | | 0.28 | | -7.90 |
| | | 44% | 0.86 | 0.84 | 13.50 | 14.21 | 0.89 | -0.31 | -7.49 | -8.86 |
| | | **81%** | | **0.78** | | **17.64** | | **-3.72** | | **-17.17** |
| QTP9_20180125 | 42% | 19% | | 0.90 | | 10.51 | | 0.89 | | -3.14 |
| | | 49% | 0.89 | 0.87 | 10.56 | 11.34 | 0.93 | 0.65 | -3.16 | -3.64 |
| | | **80%** | | **0.54** | | **19.35** | | **-9.59** | | **-25.03** |
| NX2_20180213 | 83% | 18% | | 0.94 | | 20.84 | | 18.64 | | 9.71 |
| | | 48% | 0.95 | 0.94 | 20.47 | 21.26 | 18.27 | 18.86 | 9.00 | 9.15 |
| | | 75% | | 0.93 | | 22.24 | | 19.69 | | 8.18 |

The quantitative results indicate that the recovery effectiveness of STAR typically declines significantly when cloud coverage is greater than 80%. As a result, STAR can completely remove clouds with little loss of accuracy. Only in the *NC4_20180318* scene, high overestimation occurs when cloud coverage reaches 55%. The phenomenon is caused by high cloud coverage and rapid snow variation in space and time. Therefore, users are recommended to refer to the QA maps of

430 STAR NDSI collection during snow accumulation and ablation periods, in which Bit 7 reflects the cloud coverage of the space-time block.

## 5 Data availability

The improved cloud-free Terra–Aqua MODIS NDSI collection (STAR NDSI collection) for China from 1 August 2000 to 31 July 2020, including STAR NDSI and STAR QA data, is available for download at https://doi.org/10.5281/zenodo.5644386

(Jing et al., 2021). The dataset is provided using a WGS 84 / UTM zone 48N projection, with a tag image file format (TIFF). Users can discuss and respond to issues that arise during the use of this dataset. New versions can be released in consideration of user comments. In addition, a source code for this collection is available at https://doi.org/10.5281/zenodo.6396149 (Jing, 2022).

## 6 Conclusions

STAR NDSI collection is derived from Terra–Aqua MODIS NDSI datasets using an optimized STAR from our last research (Jing et al., 2019). The evaluation tests indicate that STAR NDSI collection is highly consistent with the in-situ snow depth measurements and higher resolution NDSI maps. STAR NDSI collection generally has the following strengths. (1) This collection has reached a continuous 20-year period, which is the minimum period of a dataset for long-term hydrological and climatic processes analysis. (2) The cloud-free collection can accurately estimate the snow dynamics, highly consistent with

in-situ snow depth and Landsat NDSI maps. Specifically, STAR NDSI collection eliminates cloud contamination and preeminently improves the overall performance of TAC NDSI dataset. Due to the higher spatial resolution and larger dynamic range, the classification accuracy of STAR NDSI collection is higher than that of NIEER AVHHR SCE dataset. In terms of numerical accuracy, it is superior to MODIS CGF NDSI dataset, since the spatio-temporal adaptive fusion method generally outperforms the simplified MDC method. Additionally, it has a satisfactory accuracy in original cloud-cover areas. (3) The

collection provides a detailed snow cover dataset for China, accurately reflecting the snow conditions of the following three major snow areas: NX, NC, and QTP. The collection is available at: https://doi.org/10.5281/zenodo.5644386 (Jing et al., 2021).

As discussed above, STAR NDSI collection still has some deficiencies. A future release should consider several issues: (1) the original accuracy of MODIS NDSI datasets reduced by factors such as complex climatic conditions and dense forest coverage; (2) the reconstruction accuracy of snow edges affected by mixed pixels and high cloud coverage; (3) the

reconstruction accuracy of transient snow areas due to the inadequate spatio-temporal contextual information; and (4) the lack of evaluation based on in-situ snow depth measurements in NC due to the limited access to climate station data.

Despite the aforementioned deficiencies, since snow is a pivotal driver and sensitive indicator for many hydrometeorological processes, the daily 500 m STAR NDSI collection for 20 years has various potential applications: (1) achieving a deep understanding of long-term snow cover variability in China, (2) providing effective forcing data for hydrological and meteorological models, and (3) supporting strategic decisions on water resources management, environmental pollution governance, and related economic development.

**Author contributions.** All authors designed the methodology. YJ implemented the experiments. YJ maintained and refined STAR NDSI collection. YJ drafted the manuscript. XL and HS revised the whole manuscript. All authors provided suggestions for this manuscript.

**Competing interests.** The authors declare that they have no conflict of interest.

**Acknowledgements.** This research was supported by the National Natural Science Foundation of China (NSFC) under Grant No. 41701394. Thanks to Liupeng Lin for his constructive suggestions.

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

**Appendix A**

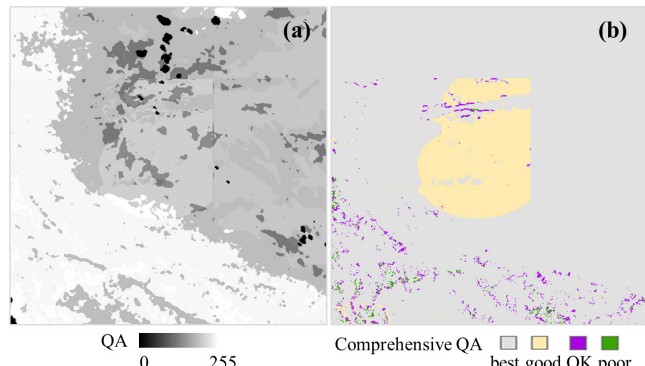
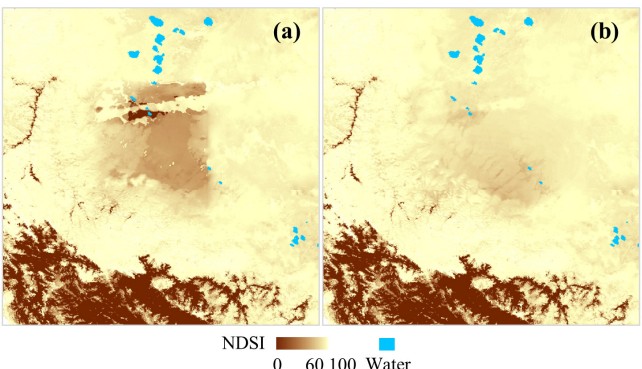

**Figure A1. QA maps over Taklimakan Desert on 19 January 2008. (a) QA map. (b) Comprehensive QA map.**

**Figure A2. Post-processing over Taklimakan Desert on 19 January 2008. (a) STAR result. (b) Final result.**

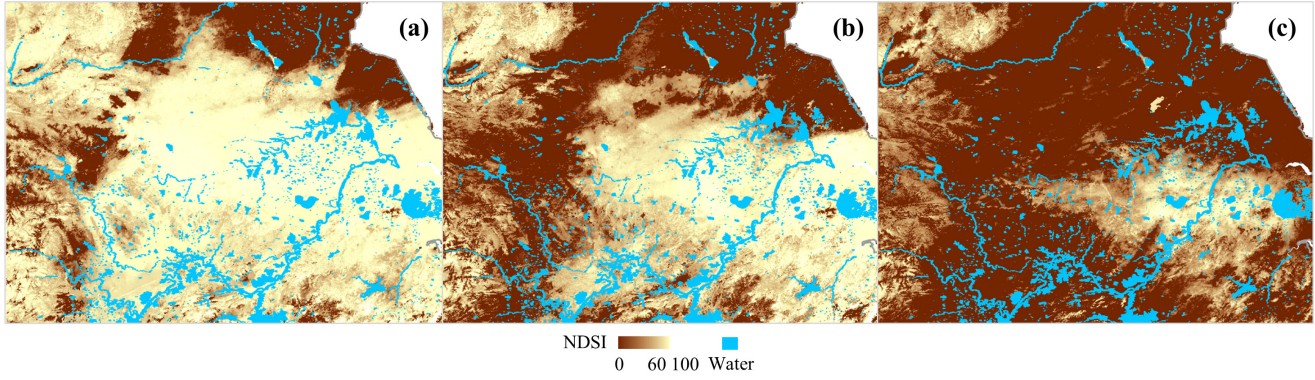

NDSI
0    60 100  Water

**Figure A3. Extreme snow event in southern China. (a) 31 January 2008. (b) 5 February 2008. (c) 10 February 2008.**