# Peer review of "STAR NDSI collection: A cloud-free MODIS NDSI dataset (2001–2020) for China"

_Earth System Science Data, 2021_

## Referee Comment (RC3)

Jing's paper produced a daily cloud-free Normalized Difference Snow Index (NDSI) product with 500 m spatial resolution based on MODIS C6 snow cover datasets in China. So far as we know, the NDSI threshold is the crucial parameter for snow detection by use of optical remote sensing data. The paper in its current version needs major revision and resubmission to meet the level expected of ESSD, for the following reasons:

Firstly, the importance of NDSI needs to be clarified in introduction and using data. NDSI is different form NDVI. The readers are more concerned about binary snow cover or fractional snow cover than NDSI itself. Therefore, it is difficult for me to evaluate whether this dataset is uniqueness or usefulness. Secondly, the current validation scheme is insufficient to support the Spatio-Temporal Adaptive fusion method.

The two issues must be addressed for this dataset to be published on ESSD.

General comments
1. The importance of the NDSI research is insufficiently described, and why NDSI is more important than binary and FSC products should be further described in the introduction.
2. The NDSI value in either MODIS C5 or C6 is the NDSI without atmospheric correction. How this NDSI differs from NDSI by the atmospheric corrected from MOD09GA/MYD09GA? Has the author compared it, and which NDSI value is more useful to readers?
3. The current validation plan (in-situ snow depth observations and Landsat NDSI maps) is insufficient to support the Spatio-Temporal Adaptive fusion method. Please add the improved validation plan.

Minor comments
1  L 95. "The daily snow cover datasets of C6 were used in this study." There are NDSI_Snow_Cover and NDSI scientific data sets in the C6 by MODIS C6 User Guide (Riggs, 2015). The NDSI_Snow_Cover and NDSI is different, the author need to describe the data used in the study. This is related to the subsequent results.
2  Fig.1. It is recommended to remove the NC snow area cover. This is only an administrative division rather than a snow region (https://essd.copernicus.org/articles/13/4711/2021/). The in-situ observations of this area were not used in this study. In addition, TP suggests replacing by QTP?
3  What does the dashed half-frame line in Fig.2 mean?
4  L 118. The description of fusion method and rules is not clear, only the priority is determined at L 123. It's better to describe the fusion method first, and then introduced the interpolation used by Aqua.
5  L158. What does NDSIP mean?
6  L 216. What does "snow-clad pixels" mean? Are there any reference?
7  L 214. Section 3.1 The validation method need to be improved.
   The in situ snow depth derived from 49 and 92 CMA station from BJ and QTP. However, the validation date need to be clear. Due to snow-free period is long,

many stations record no snow in one year. In fact, the most useful and most concerned should be the NDSI recovery during the snow cover period. The author should focus on the NDSI recovery during the snow cover period and a detailed confusion matrix needs to be given. In addition, the authors need to focus on the accuracy comparison of the product itself (TAC, L 218, "the cloud-covered areas in the TAC NDSI dataset are considered to be snow-free". Here the cloud-covered areas should be eliminated without comparison) and the final spatial continuous product (STAR). The reader is concerned with the loss of NDSI accuracy after STAR interpolation.

8  L 248. Section 3.2. The validation method need to be improved.

The focus of validation in the study should be whether the STAR method is reliable. Therefore, a reasonable verification scheme is to select actually cloud-free the Landsat NDSI maps as a reference maps, then artificially set a random 20%, 50% or 80% cloud cover (only my suggestion) on the corresponding MODIS data. The different cloud ratio maps were recovery after STAR interpolation and validated by reference Landsat NDSI maps, and the conclusions is convincing.

---

## Author Comment (AC1)

**Response to Referee #1 Comments**

**General Comment:**

The Normalized Difference Snow Index (NDSI) is vital in snow cover extent, snow cover fraction, and snow depth retrieving in case of using optical satellite observations. In this study, the authors developed an integrated cloud-free MODIS NDSI over China with the help of a spatio-temporal Adaptive fusion method. However, the following global and regional cloud-free gap-filled NDSI and snow cover extent dataset are excluded in this study, neither in the introduction section, nor in the cross-comparison section. Therefore, it is difficult for me to evaluate whether this dataset is uniqueness or usefulness.

The existing cloud-free NDSI dataset and snow cover extent datasets including:

- MODIS/Terra CGF Snow Cover Daily L3 Global 500m SIN Grid, Version 61 (https://doi.org/10.5067/MODIS/MOD10A1F.061), which provide a cloud-gap-filled daily MODIS NDSI dataset at 500m spatial resolution.
- The NIEER AVHRR snow cover extent product over China (https://essd.copernicus.org/articles/13/4711/2021/, https://doi.org/10.5194/essd-13-4711-2021), which provide a cloud-gap-filled daily AVHRR snow cover extent dataset over China.
- The daily MODIS 500m snow cover extent over China (http://data.casnw.net/portal/metadata/be3a4134-2e5c-467f-8a5e-b1c0ed6cc341, doi:10.12072/ncdc.I-SNOW.db0001.2020)
- Daily fractional snow cover dataset over High Asia during 2002 to 2018 (http://www.ncdc.ac.cn/portal/metadata/0e277d66-d89b-4e54-8a75-fe22fcc3adee, doi:10.11922/sciencedb.457)

Although the study may content material worthy of publication, the paper in its current version needs major revision and resubmission to meet the level expected of ESSD, for the following reasons.

**General Response:**

Thank you very much for the critical comments and suggestions regarding our article. Considering all the constructive comments, we carefully revised the Introduction and supplemented the cross-comparison experiments. We searched various existing cloud-free snow cover products covering China and added descriptions in the Introduction. Moreover, to elaborate the reliability of our STAR NDSI collection, we performed comparative experiments between **four existing snow cover products** and ours, all of which revealed the superiority of our

collection. All quantitative evaluation results are presented in Appendix A of the response. However, due to space limitations, we only added cross-comparisons with two typical datasets (NIEER AVHRR SCE and MODIS CGF NDSI products, considering spatial coverage and unrestricted access of datasets) in the manuscript.

In summary, the main points of the revisions include: (1) Typical long-term cloud-free snow cover products covering most snow-dominated regions in China were described in Introduction and listed in Table 1 (Section 1). (2) The classification accuracy of STAR NDSI collection was compared with those of TAC NDSI and NIEER AVHRR SCE datasets based on in-situ snow depth measurements (Section 3.1). (3) The numerical accuracy of STAR NDSI collection was compared with those of TAC NDSI datasets based on Landsat NDSI maps (Section 3.2).

Q1. Literature review in Introduction disregards former studies on NDSI retrievals from satellite data, including both datasets retrieval methods. Thus, the contribution of the current study in accordance to existing knowledge and methods is not clear. Please add the above listed cloud-free NDSI dataset and snow cover extent datasets in the introduction section.

**Response:** Thank you for the comment. We added the descriptions of various existing cloud-free snow cover products in the Introduction as follows:

...On the national scale, Huang et al. (2016) obtained a long-term cloud-removed SCA product using a multisource data fusion method. Despite many relevant studies, only a few cloud-free snow cover datasets have been released publicly.

Several typical long-term cloud-free snow cover products available online are listed in Table 1 (datasets are referenced via DOI), which cover most snow-dominated regions in China. Huang provided MODIS daily cloudless SCA products with relatively accurate snow detection capabilities in Northern Hemisphere based on multi-source data. Muhammad and Thapa (2021) obtained a daily MODIS SCA and glacier composite dataset for High Mountain Asia by aggregating seasonal, temporal, and spatial filters, which can serve as a valuable input for hydrological and glaciological investigations. Hao et al. (2021) yielded two long-term daily SCA datasets over China through a series of processes such as quality control, cloud detection, snow discrimination, and gap-filling (including hidden Markov random field and snow-depth interpolation techniques). Their releases and updates promoted the research of snow cover characteristics in China. Qiu et al. yielded a daily FSC dataset with detailed snow cover information over High Mountain Asia with MDC and spatial filtering. Additionally, the global cloud-gap-filled MODIS NDSI dataset (MOD10A1F) is available online since 2020, where cloud-covered grids in the MODIS Terra NDSI product are filled by retaining clear-sky observations from previous days (Hall and Riggs, 2020). However, this dataset

performs poorly in China, where periodic and transient snow is dominant. In general, cloud-free SCA datasets produced by composite algorithms are frequently released, while high-quality cloud-free NDSI datasets are still scarce.

To this end, this study generates a spatiotemporally continuous NDSI product with satisfactory accuracy for China, fully considering the spatio-temporal characteristics of regional snow cover variability. A Spatio-Temporal Adaptive fusion method with erroR correction (STAR) improved from our previous work (Jing et al., 2019) is utilized to eliminate cloud obscuration...

Table 1. Typical long-term cloud-free snow cover products covering most snow-dominated regions in China.

| References                | Туре | Spatial coverage    | Temporal coverage | Temporal resolution | Spatial resolution | DOI                              |
|---------------------------|------|---------------------|-------------------|---------------------|--------------------|----------------------------------|
| Hao et al. (2020)         | SCA  | China               | 2000-2020         | Daily               | ~500 m             | 10.12072/ncdc.I-SNOW.db0001.2020 |
| Hao et al. (2021)         | SCA  | China               | 1981–2019         | Daily               | ~5 km              | 10.11888/Snow.tpdc.271381        |
| Huang (2020)              | SCA  | Northern hemisphere | 2000-2015         | Daily               | ~1 km              | 10.12072/ncdc.CCI.db0044.2020    |
| Muhammad and Thapa (2021) | SCA  | High Mountain Asia  | 2002-2019         | Daily               | ~500 m             | 10.1594/PANGAEA.918198           |
| Qiu et al. (2017)*        | FSC  | High Mountain Asia  | 2002-2018         | Daily               | ~500 m             | 10.11922/sciencedb.457           |
| Hall and Riggs (2020)     | NDSI | Global coverage     | 2000-present      | Daily               | ~500 m             | 10.5067/MODIS/MOD10A1F.061       |

\*Cloud coverage is less than 10%.

Q2. The lack of innovation in accordance to existing knowledge. Please add the comparison between NDSI dataset in present study and MODIS/Terra CGF Snow Cover Daily L3 Global 500m SIN Grid, Version 61, both in fusion method and results.

Response: Thank you for the critical comment. The innovations of this work can be summarized as follows:

For the methodology, the clouds in original snow cover products are generally removed by a multi-step combination method. An accumulated cloud-free result can be obtained by combining the spatial methods and temporal methods alternately. However, the independent and successive utilization cannot take full consideration of the spatio-temporal information. Therefore, STAR NDSI collection is generated by a novel Spatio-Temporal Adaptive fusion method with erroR correction (STAR), which consists of space partition, adaptive space-time block determination, Gaussian kernel function-based fusion, and error correction. This method comprehensively considers spatial and temporal contextual information and thus is promising for the cloud removal of NDSI products. From a product perspective, cloud-free SCA datasets produced by composite algorithms are frequently released, while high-quality cloud-free NDSI datasets are still scarce. NDSI datasets can provide detailed characteristics of snow cover, which can effectively reduce the confusion of mixed pixels in SCA products. Therefore, the release of cloud-free NDSI datasets with high accuracy is of great significance.

In addition, to elaborate the accuracy of our STAR NDSI collection, we added two cross-comparisons with two typical snow cover datasets including NIEER AVHRR SCE (Hao et al., 2021) and MODIS CGF NDSI (Hall and Riggs, 2020) products. The main revisions are as follows:

**3** Results**

As mentioned above, the generation procedure of continuous snow collection includes the pre-process TAC and the key-process STAR. The remainder clouds of 30.62% in the entire collection after TAC are completely removed by STAR. To elaborate the reliability of STAR NDSI collection, TAC NDSI, NIEER AVHRR SCE, and MODIS CGF NDSI products are used as baseline data. Specifically, based on in-situ snow depth measurements, the classification accuracy of STAR NDSI collection is compared with those of TAC NDSI and NIEER AVHRR SCE datasets. In addition, based on Landsat NDSI maps, its numerical accuracy is compared with those of TAC NDSI and NIEER AVHRR SCE and MODIS CGF NDSI datasets. This section presents the evaluation results, followed by an exemplary application.

**3.1 Validation against in-situ snow depth measurements**

As described above, the in-situ snow depth data in XJ from 1 January 2001 to 31 August 2007 and on TP from 1 August 2000 to 31 December 2013, were used as the ground truth to evaluate the classification accuracy of TAC NDSI, NIEER AVHRR SCE, and STAR NDSI datasets. The nearest pixel was matched with each meteorological station, with a total of about 600000 data pairs. Snow-clad pixels in NDSI datasets range from 10 to 100, whereas snow-free pixels are 0; thus, the classification threshold is set as 10 (Zhang et al., 2019). The discriminant threshold for in-situ snow depth is set as 0 or 1 cm. In addition, the cloud-covered areas in TAC NDSI dataset are considered to be snow-free.

Table 4 demonstrates that NIEER AVHRR SCE and STAR NDSI datasets preeminently capture the snow dynamics in XJ referring to the in-situ measurements, with OAs more than 90%. However, TAC NDSI dataset is insufficient to accurately describe the snow cover variability. Although CEs perform well regardless of the snow depth threshold, OEs of TAC NDSI collection are extremely high, indicating that many cloud-covered areas are dominated by snow. NIEER AVHRR SCE dataset partially retrieves snow pixel under cloud obstruction with an OE decreased by ~43%. STAR NDSI collection completely removes clouds and accurately presents snow distribution, with an OE further decreased from ~17% to ~7%. The generation procedure in XJ has two strengths. Firstly, the satellite-borne sensors can accurately capture the snow events on the ground due to the generally thick snow averaging approximately 20 cm. Secondly, the gap-filling approach with comprehensive consideration of spatial and temporal correlation has outstanding reliability due to the significant periodicity of snow variation. It can be inferred that the NDSI datasets in NC have high accuracy because of the similar snow conditions, despite

the lack of in-situ data in this region.

By contrast, despite the satisfactory performance of OAs and CEs, the OEs of three snow cover datasets over TP are as remarkably high as 72%, 40%, and 39% even at the snow depth threshold of 1 cm (Table 5). This finding indicates the omission of a large number of snow-covered pixels. The specific reasons are as follows. Firstly, the original MODIS NDSI maps frequently underestimate the snow presence throughout the snow period because discriminating the shallow snow pixels with an averaged snow depth of approximately 4 cm over TP is challenging. Secondly, the credibility of the spatio-temporal contextual information is relatively low because the snow rapidly and irregularly varies due to the extremely complex topographic and climatic conditions, leading to a further decrease in the accuracy of the gap-filled results. Lastly, the meteorological stations over TP are unevenly distributed and are mostly located in low- and medium-altitude/latitude areas dominated by transient snow. Consequently, the evaluation results slightly exaggerate the real OEs.

Overall, STAR NDSI collection is capable of snow status estimation, eliminating cloud contamination in TAC NDSI dataset, and capturing more snow events than NIEER AVHRR SCE dataset. However, the accuracy of STAR NDSI collection has a significant regional heterogeneity. On the one hand, the accuracy over TP is lower than that of XJ and NC, which is consistent with the characteristic of the original MODIS NDSI maps. On the other hand, the permanent and periodic snow regime regions reconstructed by STAR have prominently high accuracy, while the transient snow-covered regions are easily omitted. Fortunately, the monitoring of permanent and periodic snow plays a key role in most snow-related investigations.

| Table 4. | Classification | statistics                              | based on | in XJ. |
|----------|----------------|-----------------------------------------|----------|--------|
|          |                | ~ ~ ~ ~ ~ ~ ~ ~ ~ ~ ~ ~ ~ ~ ~ ~ ~ ~ ~ ~ |          |        |

Table 5. Classification statistics over TP.

|            | Sno                   | w depth > | • 0 cm | Snow depth $> 1$ cm |                       |      |    | Sno  | w depth >  | • 0 cm | Snow depth $> 1$ cm  |       |      |
|------------|-----------------------|-----------|--------|---------------------|-----------------------|------|----|------|------------|--------|----------------------|-------|------|
| Indicators | (Snow fraction = 30%) |           |        | (Snov               | (Snow fraction = 28%) |      |    | (Sno | w fraction | = 5%)  | (Snow fraction = 3%) |       |      |
|            | TAC                   | NIEER     | STAR   | TAC                 | NIEER                 | STAR |    | TAC  | NIEER      | STAR   | TAC                  | NIEER | STAR |
| OA         | 0.81                  | 0.94      | 0.95   | 0.82                | 0.94                  | 0.95 | OA | 0.94 | 0.93       | 0.95   | 0.96                 | 0.94  | 0.96 |
| CE         | 0.02                  | 0.02      | 0.04   | 0.02                | 0.03                  | 0.05 | CE | 0.01 | 0.04       | 0.02   | 0.02                 | 0.05  | 0.03 |
| OE         | 0.60                  | 0.17      | 0.07   | 0.58                | 0.15                  | 0.05 | OE | 0.78 | 0.53       | 0.52   | 0.72                 | 0.42  | 0.39 |

Note: TAC, NIEER, and STAR represent TAC NDSI, NIEER AVHRR SCE, and STAR NDSI datasets, respectively.

**3.2 Validation based on Landsat NDSI maps**

...Two evaluations including a cross-comparison of TAC NDSI, MODIS CGF NDSI, and STAR NDSI datasets and an internal comparison of clear-sky and cloud-cover areas are described in detail below.

For the cross-comparison, the visual effects of three NDSI datasets on 8 January 2018 and 3 February 2018 are shown in Fig. 4. TAC NDSI dataset is still heavily obscured by clouds. Although MODIS CGF NDSI dataset completely removes clouds from the MOD10A1 product, it is difficult to accurately retrieve periodic and transient

snow cover areas due to the simplicity of the cloud-gap-filled method. Specifically, the gaps are filled by retaining clear-sky observations from previous days. However, snow patterns under cloud cover are likely to change significantly during these days. Therefore, snow cover is significantly underestimated during accumulation (Fig. 4, a2) and overestimated during ablation (Fig. 4, b2). By contrast, STAR NDSI collection preeminently captures the snow dynamics under temporally continuous clouds, attributing to the spatio-temporal adaptive fusion strategy. Furthermore, the three NDSI datasets are quantitatively assessed by Landsat NDSI maps.

---

## Author Comment (AC2)

**Response to Referee #2 Comments**

**General Comment:**

The manuscript titled "STAR NDSI collection: A cloud-free MODIS NDSI dataset (2001–2020) for China" estimates cloud-free snow data for China. The authors use Spatio-Temporal Adaptive fusion method with erroR correction (STAR) to derive snow cover. Cloud cover is the main obstacle in passive remote sensing snow monitoring and is important to overcome. The study is important but there are few major issues in the present form which needs to be addressed.

**General Response:**

Thank you very much for the critical comments and suggestions regarding our article. Considering all the constructive comments, we carefully revised the manuscript and supplemented a comparative experiment. Based on the experimental results, the priority scheme of the pre-process TAC and the accuracy of STAR NDSI collection were discussed in detail.

In summary, the main points of the response include: (1) Combined with the comments of referee #1, we searched various existing cloud-free snow cover products covering China and added descriptions in the Introduction. (2) A comparative experiment between the modified Terra and Aqua combination (TAC) and the original TAC was performed. (3) The uncertainty caused by the pre-process TAC and the accuracy of STAR NDSI collection were analyzed. (4) The proposed cloud removal method was re-emphasized. (5) The code was uploaded to Zenode.

**Major comments:**

Q1. The authors use combined Terra and Aqua MODIS data in this manuscript. They combine the data first and then use STAR method consisting of spatio-temporal adaptive fusion (STAF) and error correction (EC). Combining Terra and Aqua this way potentially overestimates snow (Muhammad and Thapa, 2020, 2021). The authors are suggested to either revise the TAC or explain the potential uncertainty.

**Response:** Thank you for the critical comment. Firstly, we carefully learned the cloud removal method proposed by Muhammad and Thapa (2020, 2021). Then, we analyzed the necessity of TAC using the maximum strategy. Finally, we performed a comparative experiment between the modified TAC and the original TAC to demonstrate the analysis. The priority schemes of the modified TAC and the original TAC are determined as *low*

*value > high value > cloud* and *high value > low value > cloud*, respectively.

Muhammad and Thapa (2020, 2021) suggested merging Terra and Aqua 8-day binary snow cover products in a way that considered snow only where pixels in both the products are classified as snow (i.e., *no snow* > *snow* > *cloud*). A multi-step combination method was used to remove clouds from MODIS snow cover area (SCA) products, consisting of seasonal filtering, temporal filtering, spatial filtering, and TAC. This priority scheme of TAC is an inter-verification of Terra and Aqua 8-day snow cover products which generally overestimate the snow cover extent. It also helps to avoid uncertainty produced using spatial filtering. Therefore, it is superior to the priority scheme of *snow* > *no snow* > *cloud*. However, they also suggested that "We do not recommend this merging criteria for daily snow products in mountainous areas due to the error of omission which may be further increased because of the off-nadir-view acquisition and edge pixels".

For STAR NDSI collection, the pre-process TAC is used to combine the Terra and Aqua MODIS daily NDSI datasets. Since there is no significant overestimation in the daily original datasets and no uncertainty caused by spatial filtering, the priority scheme is set as *high value > low value > cloud*. The four quantitative evaluations in the manuscript are shown below (Revised results based on the comments of Referee #1). Compared with NIEER AVHRR SCE (Hao et al., 2021) and MODIS CGF NDSI (Hall and Riggs, 2020) products, our STAR NDSI collection presents optimal classification accuracy and numerical accuracy. All the results reveal that our STAR NDSI collection tends to slightly underestimate rather than overestimate snow cover area. The omission errors (OEs) in Table 4 and Table 5 are relatively significant and the differences in snow rate (SRDs, MODIS – Landsat) in Table 6 and Table 7 are generally negative. These findings are in line with their assessment of daily snow products (Muhammad and Thapa, 2020). Particularly, the absolute errors (AEs, MODIS – Landsat) in the evaluations based on Landsat NDSI maps are generally positive. This is mainly due to the different spectral response curves of the bands on different sensors.

**3.1 Validation against in-situ snow depth measurements**

| Table 4. Classification statistics based on in A. | Table 4. | Classification | statistics | based | on in | ιXJ |
|---------------------------------------------------|----------|----------------|------------|-------|-------|-----|
|---------------------------------------------------|----------|----------------|------------|-------|-------|-----|

Table 5. Classification statistics over TP.

|            | Sno   | w depth >  | • 0 cm | Sno   | w depth >  | · 1 cm |            | Sno   | w depth >  | 0 cm  | Sno  | w depth >  | 1 cm  |
|------------|-------|------------|--------|-------|------------|--------|------------|-------|------------|-------|------|------------|-------|
| Indicators | (Snov | v fraction | = 30%) | (Snow | v fraction | = 28%) | Indicators | (Snor | w fraction | = 5%) | (Sno | w fraction | = 3%) |
|            | TAC   | NIEER      | STAR   | TAC   | NIEER      | STAR   |            | TAC   | NIEER      | STAR  | TAC  | NIEER      | STAR  |
| OA         | 0.81  | 0.94       | 0.95   | 0.82  | 0.94       | 0.95   | OA         | 0.94  | 0.93       | 0.95  | 0.96 | 0.94       | 0.96  |
| CE         | 0.02  | 0.02       | 0.04   | 0.02  | 0.03       | 0.05   | CE         | 0.01  | 0.04       | 0.02  | 0.02 | 0.05       | 0.03  |
| OE         | 0.60  | 0.17       | 0.07   | 0.58  | 0.15       | 0.05   | OE         | 0.78  | 0.53       | 0.52  | 0.72 | 0.42       | 0.39  |

Note: TAC, NIEER, and STAR represent TAC NDSI, NIEER AVHRR SCE, and STAR NDSI datasets, respectively.

**3.2 Validation based on Landsat NDSI maps**

[revised manuscript text omitted]

A comparative experiment was performed to further demonstrate the effectiveness of the original TAC, as shown in Supplementary Table 1 (not added to the manuscript). In this experiment, the priority schemes of the modified TAC and the original TAC are determined as *low value > high value > cloud* and *high value > low value > cloud*, respectively. Combining Terra and Aqua daily NDSI products with a minimum strategy, the average AE and RMSE are slightly decreased by 1.8 and 0.21, respectively. However, the SRDs are worsened compared with Landsat NDSI maps, especially on Tibetan Plateau (TP). The OEs are also expected to be worsened compared with in-situ snow depth measurements. In addition, this underestimation of snow cover area will be further amplified after the key-process STAR due to less snow cover information. As a result, we also do not recommend the pre-process TAC with a minimum strategy in the cloud removal of daily MODIS NDSI products.

Supplementary Table 1. Performance statistics for the modified TAC (TAC\_MIN) and the original TAC (TAC\_MAX) in clear-sky areas according to the TAC dataset. Note: Red bold values indicate a significant degradation of the modified TAC compared with the original TAC used for STAR NDSI collection. Purple and bold Regions/Dates are the newly added Regions/Dates on TP.

| Bagion Data   | С       | С       | RM      | SE      | A       | E       | SRD     | (%)     |
|---------------|---------|---------|---------|---------|---------|---------|---------|---------|
| Region_Date   | TAC_MAX | TAC_MIN | TAC_MAX | TAC_MIN | TAC_MAX | TAC_MIN | TAC_MAX | TAC_MIN |
| NC1_20180225  | 0.95    | 0.95    | 16.76   | 16.44   | 15.17   | 14.27   | 2.91    | 0.17    |
| NC2_20180311  | 0.90    | 0.90    | 12.21   | 12.20   | 11.75   | 11.74   | -0.08   | -0.08   |
| NC3_20180311  | 0.92    | 0.92    | 2.34    | 2.34    | -0.09   | -0.12   | -1.03   | -1.24   |
| NC4_20180318  | 0.98    | 0.98    | 10.34   | 9.52    | 4.90    | 4.11    | -1.20   | -1.99   |
| CCR1_20180203 | 0.83    | 0.88    | 3.35    | 2.37    | 0.71    | 0.09    | 3.67    | -0.25   |
| CCR2_20180203 | 0.54    | 0.56    | 11.00   | 10.40   | 6.90    | 6.34    | 33.43   | 31.59   |
| TP1_20180322  | 0.75    | 0.76    | 10.26   | 9.43    | 0.51    | -1.01   | 0.51    | -4.07   |
| TP2_20180225  | 0.86    | 0.82    | 13.50   | 15.97   | 0.86    | -4.84   | -7.51   | -18.59  |
| TP3_20180320  | 0.75    | 0.75    | 3.69    | 3.77    | -0.40   | -0.57   | -1.10   | -1.93   |
| TP4_20180401  | 0.80    | 0.69    | 16.31   | 22.89   | -4.40   | -11.44  | -8.42   | -20.90  |
| TP5_20180307  | 0.94    | 0.94    | 13.57   | 12.94   | 8.21    | 6.26    | 1.83    | -1.97   |
| TP6_20180305  | 0.80    | 0.76    | 14.87   | 16.97   | 1.79    | -0.03   | -4.75   | -8.12   |
| TP7_20180107  | 0.98    | 0.98    | 15.13   | 14.51   | 10.27   | 9.16    | -0.93   | -3.81   |
| TP8_20180128  | 0.76    | 0.82    | 10.58   | 7.96    | 3.12    | 1.48    | 7.51    | 2.36    |
| XJ1_20180105  | 0.89    | 0.85    | 24.47   | 24.07   | 24.28   | 23.88   | 0.00    | 0.00    |
| XJ2_20180213  | 0.95    | 0.95    | 20.53   | 18.46   | 18.35   | 16.35   | 9.07    | 8.05    |

| Average       | 0.07 | 0.95 | 12 00 | 12 77 | 6 27  | 4 47  | 0.96  | 2 20          |
|---------------|------|------|-------|-------|-------|-------|-------|---------------|
| TP14_20180226 | 0.82 | 0.81 | 12.11 | 12.76 | -4.72 | -5.41 | -5.73 | -8.6 7 |
| TP13_20180307 | 0.94 | 0.94 | 12.12 | 12.12 | 5.30  | 4.61  | -1.76 | -4.63         |
| TP12_20180312 | 0.97 | 0.97 | 10.24 | 8.86  | 5.11  | 2.85  | 1.13  | -0.41         |
| TP11_20180107 | 0.93 | 0.94 | 12.14 | 11.70 | 4.74  | 3.93  | -0.79 | -5.08         |
| TP10_20180305 | 0.83 | 0.80 | 12.97 | 14.15 | -3.44 | -5.08 | -5.73 | -10.40        |
| TP9_20180125  | 0.90 | 0.84 | 10.35 | 11.96 | 0.93  | -3.29 | -3.16 | -12.33        |
| XJ5_20180220  | 0.98 | 0.97 | 8.63  | 7.75  | 3.15  | 2.46  | -1.24 | -1.79         |
| XJ4_20180103  | 0.67 | 0.65 | 29.53 | 23.54 | 28.18 | 22.23 | 0.62  | 0.56          |
| XJ3_20180220  | 0.93 | 0.91 | 17.49 | 16.21 | 15.50 | 13.90 | 4.15  | 3.74          |

Q2. The authors missed to share the code to generate STAR NDSI dataset. It is incomplete without sharing the code. The code is also required to evaluate the methodology as well.

**Response:** Thank you for the critical comment. We uploaded the code to Zenode and added a description in the Data availability as follows. We promise to update Closed Access to Open Access when the manuscript is accepted.

The improved cloud-free MODIS NDSI collection (STAR NDSI collection) for China from 1 August 2000 to 31 July 2020, including STAR NDSI and STAR QA data, is available for download at https://doi.org/10.5281/zenodo.5644386 (Jing et al., 2021). The dataset is provided using a WGS 84 / UTM zone 48N projection, with a tag image file format (TIFF). Users can discuss and respond to issues that arise during the use of this dataset. New versions can be released in consideration of user comments. **In addition, a source code for this collection is available at https://doi.org/10.5281/zenodo.6396149** (Jing, 2022).

Q3. The C6 snow is in NDSI ranging between 0 and 100. It is not explained how the authors reconstructed the snow data. It is a challenge to improve the data on how to replace the cloudy pixel, so it is significant to understand the way the value is replaced.

**Response:** Thank you for the comment. A Spatio-Temporal Adaptive fusion method with erroR correction (STAR) improved from our previous work (Jing et al., 2019) is utilized to eliminate cloud obscuration. This cloud removal method is detailed in the Algorithm description (Section 2.2).

For the methodology, the clouds are generally removed by a multi-step combination method in existing snow cover products. An accumulated cloud-free result can be obtained by combining the spatial methods and temporal methods alternately. However, the independent and successive utilization cannot take full consideration of the spatio-temporal information. Therefore, STAR NDSI collection is generated by a novel Spatio-Temporal Adaptive fusion method with erroR correction (STAR), which consists of space partition, adaptive space-time block determination, Gaussian kernel function-based fusion, and error correction. This method comprehensively

considers spatial and temporal contextual information and thus is promising for the cloud removal of NDSI products. The cloud removal procedure is described as follows.

**2.2 Algorithm description**

MODIS NDSI datasets are unable to represent the daily conditions of snow accumulation and ablation accurately because the optical remote-sensed images are subject to severe cloud pollution. Therefore, a Spatio-Temporal Adaptive fusion method with erroR correction (STAR), which is derived from our two-stage spatio-temporal fusion method (Jing et al., 2019), is presented to produce a spatio-temporal continuous snow collection. As shown in Fig. 2, the generation procedure comprises the pre-process TAC and the key-process STAR. Then, a quality assessment (QA) approach is presented to provide a data reliability profile for users. On this basis, post-processing is used to further improve the data quality in individual abnormal areas.

Figure 2. Schematic of the generation procedure of STAR NDSI collection.

**2.2.1 Terra and Aqua combination (TAC)**

TAC blends the same-day snow maps deriving from MODIS sensors onboard Terra and Aqua satellites. Its cornerstone is the unlikely significant changes of the snow pattern within the data-acquired time interval (approximately 3 h). The improved Aqua MODIS C6 NDSI dataset significantly enhances the effectiveness of TAC due to the successful restoration of the absent Aqua MODIS band 6 data by the quantitative image restoration method (Gladkova et al., 2012). TAC can efficiently decrease the cloud fraction by 5%–20% with negligible precision sacrifice (Li et al., 2019). Thus, this method is introduced as a pre-processing to reduce cloud coverage preliminarily. Its priority scheme is determined as high value > low value > cloud. Particularly, the snow in low altitude and low latitude areas during summer is reversed to no snow to alleviate commission errors inherited from the original data. In addition, since the Aqua dataset is available since July 2002, the key-process STAR is directly used to remove clouds from Terra MODIS NDSI dataset between August 2000 and May 2002.

**2.2.2 Spatio-Temporal Adaptive fusion with erroR correction (STAR)**

Many regions with persistent clouds are out of the scope of TAC. To this end, an advanced STAR method, which comprehensively utilizes spatio-temporal contextual information, is proposed to remove the clouds thoroughly. As shown in Fig. 3, the method performs in two passes: spatio-temporal adaptive fusion (STAF) and error correction (EC).

---

## Author Response (AR1)

**Response to Referee #1 Comments**

**General Comment:**

The Normalized Difference Snow Index (NDSI) is vital in snow cover extent, snow cover fraction, and snow depth retrieving in case of using optical satellite observations. In this study, the authors developed an integrated cloud-free MODIS NDSI over China with the help of a spatio-temporal Adaptive fusion method. However, the following global and regional cloud-free gap-filled NDSI and snow cover extent dataset are excluded in this study, neither in the introduction section, nor in the cross-comparison section. Therefore, it is difficult for me to evaluate whether this dataset is uniqueness or usefulness.

The existing cloud-free NDSI dataset and snow cover extent datasets including:

- MODIS/Terra CGF Snow Cover Daily L3 Global 500m SIN Grid, Version 61 (https://doi.org/10.5067/MODIS/MOD10A1F.061), which provide a cloud-gap-filled daily MODIS NDSI dataset at 500m spatial resolution.

- The NIEER AVHRR snow cover extent product over China (https://essd.copernicus.org/articles/13/4711/2021/, https://doi.org/10.5194/essd-13-4711-2021), which provide a cloud-gap-filled daily AVHRR snow cover extent dataset over China.

- The daily MODIS 500m snow cover extent over China (http://data.casnw.net/portal/metadata/be3a4134-2e5c-467f-8a5e-b1c0ed6cc341, doi:10.12072/ncdc.I-SNOW.db0001.2020)

- Daily fractional snow cover dataset over High Asia during 2002 to 2018 (http://www.ncdc.ac.cn/portal/metadata/0e277d66-d89b-4e54-8a75-fe22fcc3adee, doi:10.11922/sciencedb.457)

Although the study may content material worthy of publication, the paper in its current version needs major revision and resubmission to meet the level expected of ESSD, for the following reasons.

**General Response:**

Thank you very much for the critical comments and suggestions regarding our article. Considering all the constructive comments, we carefully revised the Introduction and supplemented the cross-comparison experiments. We searched various existing cloud-free snow cover products covering China and added descriptions in the Introduction. Moreover, to elaborate the reliability of our STAR NDSI collection, we performed comparative experiments between **four existing snow cover products** and ours, all of which revealed the superiority of our

collection. **All quantitative evaluation results are presented in Appendix A of the response**. However, due to space limitations, we only added cross-comparisons with two typical datasets (NIEER AVHRR SCE and MODIS CGF NDSI products, considering spatial coverage and unrestricted access of datasets) in the manuscript.

**In summary, the main points of the revisions include: (1)** Typical long-term cloud-free snow cover products covering most snow-dominated regions in China were described in Introduction and listed in Table 1 (Section 1). **(2)** The classification accuracy of STAR NDSI collection was compared with those of TAC NDSI and NIEER AVHRR SCE datasets based on in-situ snow depth measurements (Section 3.1). **(3)** The numerical accuracy of STAR NDSI collection was compared with those of TAC NDSI and MODIS CGF NDSI datasets based on Landsat NDSI maps (Section 3.2).

Q1. Literature review in Introduction disregards former studies on NDSI retrievals from satellite data, including both datasets retrieval methods. Thus, the contribution of the current study in accordance to existing knowledge and methods is not clear. Please add the above listed cloud-free NDSI dataset and snow cover extent datasets in the introduction section.

**Response:** Thank you for the comment. We added the descriptions of various existing cloud-free snow cover products in the Introduction as follows:

…On the national scale, Huang et al. (2016) obtained a long-term cloud-removed SCA product using a multi-source data fusion method. Despite many relevant studies, only a few cloud-free snow cover datasets have been released publicly.

Several typical long-term cloud-free snow cover products available online are listed in Table 1 (datasets are referenced via DOI), which cover most snow-dominated regions in China. Huang provided MODIS daily cloudless SCA products with relatively accurate snow detection capabilities in Northern Hemisphere based on multi-source data. Muhammad and Thapa (2021) obtained a daily MODIS SCA and glacier composite dataset for High Mountain Asia by aggregating seasonal, temporal, and spatial filters, which can serve as a valuable input for hydrological and glaciological investigations. Hao et al. (2021) yielded two long-term daily SCA datasets over China through a series of processes such as quality control, cloud detection, snow discrimination, and gap-filling (including hidden Markov random field and snow-depth interpolation techniques). Their releases and updates promoted the research of snow cover characteristics in China. Qiu et al. yielded a daily FSC dataset with detailed snow cover information over High Mountain Asia with MDC and spatial filtering. Additionally, the global cloud-gap-filled MODIS NDSI dataset (MOD10A1F) is available online since 2020, where cloud-covered grids in the MODIS Terra NDSI product are filled by retaining clear-sky observations from previous days (Hall and Riggs, 2020). However, this dataset

performs poorly in China, where periodic and transient snow is dominant. In general, cloud-free SCA datasets produced by composite algorithms are frequently released, while high-quality cloud-free NDSI datasets are still scarce.

To this end, this study generates a spatiotemporally continuous NDSI product with satisfactory accuracy for China, fully considering the spatio-temporal characteristics of regional snow cover variability. A Spatio-Temporal Adaptive fusion method with erroR correction (STAR) improved from our previous work (Jing et al., 2019) is utilized to eliminate cloud obscuration…

**Table 1. Typical long-term cloud-free snow cover products covering most snow-dominated regions in China.**

| References | Type | Spatial coverage | Temporal coverage | Temporal resolution | Spatial resolution | DOI |
|---|---|---|---|---|---|---|
| Hao et al. (2020) | SCA | China | 2000–2020 | Daily | ~500 m | 10.12072/ncdc.I-SNOW.db0001.2020 |
| Hao et al. (2021) | SCA | China | 1981–2019 | Daily | ~5 km | 10.11888/Snow.tpdc.271381 |
| Huang (2020) | SCA | Northern hemisphere | 2000–2015 | Daily | ~1 km | 10.12072/ncdc.CCI.db0044.2020 |
| Muhammad and Thapa (2021) | SCA | High Mountain Asia | 2002–2019 | Daily | ~500 m | 10.1594/PANGAEA.918198 |
| Qiu et al. (2017)* | FSC | High Mountain Asia | 2002–2018 | Daily | ~500 m | 10.11922/sciencedb.457 |
| Hall and Riggs (2020) | NDSI | Global coverage | 2000–present | Daily | ~500 m | 10.5067/MODIS/MOD10A1F.061 |

*Cloud coverage is less than 10%.

Q2. The lack of innovation in accordance to existing knowledge. Please add the comparison between NDSI dataset in present study and MODIS/Terra CGF Snow Cover Daily L3 Global 500m SIN Grid, Version 61, both in fusion method and results.

**Response:** Thank you for the critical comment. The innovations of this work can be summarized as follows:

**For the methodology**, the clouds in original snow cover products are generally removed by a multi-step combination method. An accumulated cloud-free result can be obtained by combining the spatial methods and temporal methods alternately. However, the independent and successive utilization cannot take full consideration of the spatio-temporal information. Therefore, STAR NDSI collection is generated by a novel Spatio-Temporal Adaptive fusion method with erroR correction (STAR), which consists of space partition, adaptive space-time block determination, Gaussian kernel function-based fusion, and error correction. This method comprehensively considers spatial and temporal contextual information and thus is promising for the cloud removal of NDSI products. **From a product perspective**, cloud-free SCA datasets produced by composite algorithms are frequently released, while high-quality cloud-free NDSI datasets are still scarce. NDSI datasets can provide detailed characteristics of snow cover, which can effectively reduce the confusion of mixed pixels in SCA products. Therefore, the release of cloud-free NDSI datasets with high accuracy is of great significance.

In addition, to elaborate the accuracy of our STAR NDSI collection, **we added two cross-comparisons with two typical snow cover datasets** including NIEER AVHRR SCE (Hao et al., 2021) and MODIS CGF NDSI (Hall and Riggs, 2020) products. The main revisions are as follows:

[revised manuscript text omitted]

Q3. The lack of depth in the result analysis that makes the study inconclusive. Please emphasize the unique contributions in the present study in the comparison with the above listed cloud-free NDSI dataset and snow cover extent datasets over China.

**Response:** Thank you very much for the suggestion. More rigorous evaluation and more detailed analysis were added to the Results (Section 3), as shown in the response to Q2. Furthermore, we re-emphasized our contributions in the Conclusions (Section 5). The related revisions are as follows:

[revised manuscript text omitted]

**Appendix A**

All quantitative evaluation results are presented here. Due to the different spatial and temporal coverage of various existing datasets, each dataset was compared with TAC NDSI and STAR NDSI datasets separately. Specifically, based on in-situ snow depth measurements, the classification accuracy of STAR NDSI collection was compared with three SCA/FSC products (SCA also called SCE). Based on Landsat NDSI maps, the numerical accuracy of STAR NDSI collection was compared with another NDSI product.

**1. NIEER AVHRR snow cover extent product over China (Hao et al., 2021).**

**Table A1. Classification statistics based on in XJ.**       **Table A2. Classification statistics over TP.**

| Indicators | Snow depth > 0 cm (Snow fraction = 30%) | | | Snow depth > 1 cm (Snow fraction = 28%) | | | Indicators | Snow depth > 0 cm (Snow fraction = 5%) | | | Snow depth > 1 cm (Snow fraction = 3%) | | |
|---|---|---|---|---|---|---|---|---|---|---|---|---|---|
| | TAC | **NIEER** | **STAR** | TAC | **NIEER** | **STAR** | | TAC | **NIEER** | **STAR** | TAC | **NIEER** | **STAR** |
| OA | 0.81 | **0.94** | **0.95** | 0.82 | **0.94** | **0.95** | OA | 0.94 | **0.93** | **0.95** | 0.96 | **0.94** | **0.96** |
| CE | 0.02 | **0.02** | **0.04** | 0.02 | **0.03** | **0.05** | CE | 0.01 | **0.04** | **0.02** | 0.02 | **0.05** | **0.03** |
| OE | 0.60 | **0.17** | **0.07** | 0.58 | **0.15** | **0.05** | OE | 0.78 | **0.53** | **0.52** | 0.72 | **0.42** | **0.39** |

Note: TAC, NIEER, and STAR represent TAC NDSI, NIEER AVHRR SCE, and STAR NDSI datasets, respectively.

**2. The daily MODIS 500m snow cover extent over China (Hao et al., 2020).**

**Table A3. Classification statistics based on in XJ.**       **Table A4. Classification statistics over TP.**

| Indicators | Snow depth > 0 cm (Snow fraction = 30%) | | | Snow depth > 1 cm (Snow fraction = 28%) | | | Indicators | Snow depth > 0 cm (Snow fraction = 5%) | | | Snow depth > 1 cm (Snow fraction = 3%) | | |
|---|---|---|---|---|---|---|---|---|---|---|---|---|---|
| | TAC | **NIEER1** | **STAR** | TAC | **NIEER1** | **STAR** | | TAC | **NIEER1** | **STAR** | TAC | **NIEER1** | **STAR** |
| OA | 0.81 | **0.93** | **0.95** | 0.82 | **0.93** | **0.95** | OA | 0.94 | **0.96** | **0.95** | 0.96 | **0.97** | **0.96** |
| CE | 0.02 | **0.02** | **0.04** | 0.02 | **0.03** | **0.05** | CE | 0.01 | **0.01** | **0.02** | 0.02 | **0.01** | **0.03** |
| OE | 0.60 | **0.17** | **0.07** | 0.58 | **0.15** | **0.05** | OE | 0.78 | **0.54** | **0.52** | 0.72 | **0.40** | **0.39** |

Note: TAC, NIEER1, and STAR represent TAC NDSI, NIEER MODIS SCA, and STAR NDSI datasets, respectively.

**3. Daily fractional snow cover dataset over High Asia from 2002 to 2018 (Qiu et al., 2017).**

**Table A5. Classification statistics based on in XJ.**       **Table A6. Classification statistics over TP.**

| Indicators | Snow depth > 0 cm (Snow fraction = 30%) | | | Snow depth > 1 cm (Snow fraction = 28%) | | | Indicators | Snow depth > 0 cm (Snow fraction = 5%) | | | Snow depth > 1 cm (Snow fraction = 3%) | | |
|---|---|---|---|---|---|---|---|---|---|---|---|---|---|
| | TAC | **FSC** | **STAR** | TAC | **FSC** | **STAR** | | TAC | **FSC** | **STAR** | TAC | **FSC** | **STAR** |
| OA | 0.82 | **0.79** | **0.95** | 0.83 | **0.80** | **0.95** | OA | 0.94 | **0.91** | **0.95** | 0.96 | **0.92** | **0.96** |
| CE | 0.02 | **0.04** | **0.04** | 0.02 | **0.04** | **0.05** | CE | 0.02 | **0.06** | **0.02** | 0.02 | **0.06** | **0.03** |
| OE | 0.58 | **0.71** | **0.07** | 0.56 | **0.71** | **0.05** | OE | 0.77 | **0.80** | **0.51** | 0.71 | **0.77** | **0.39** |

Note: TAC, FSC, and STAR represent TAC NDSI, HMA MODIS FSC, and STAR NDSI datasets, respectively. FSC greater than 50% is considered snow.

**4. MODIS/Terra CGF Snow Cover Daily L3 Global 500m SIN Grid, Version 61 (Hall and Riggs, 2020).**

**Table A7. Performance statistics for two MODIS NDSI datasets against Landsat NDSI maps.**

| Region_Date | Cloud cover (%) | CC | | | RMSE | | | AE | | | SRD (%) | | |
|---|---|---|---|---|---|---|---|---|---|---|---|---|---|
| | | TAC | **CGF** | **STAR** | TAC | **CGF** | **STAR** | TAC | **CGF** | **STAR** | TAC | **CGF** | **STAR** |
| NC1_20180225 | 61.4 | 0.49 | **0.90** | **0.87** | 25.64 | **20.20** | **17.10** | -11.59 | **18.35** | **15.07** | -60.1 | **1.0** | **1.0** |
| NC2_20180311 | 43.6 | 0.69 | **0.72** | **0.83** | 26.75 | **19.77** | **13.87** | -9.36 | **15.72** | **12.14** | -43.1 | **-5.7** | **-1.7** |
| NC3_20180311 | 34.6 | 0.19 | **0.86** | **0.86** | 18.61 | **20.97** | **8.79** | -10.26 | **10.78** | **0.15** | -34.2 | **0.0** | **-2.8** |
| NC4_20180318 | 16.3 | 0.73 | **0.95** | **0.98** | 21.50 | **15.03** | **10.25** | -3.33 | **8.26** | **5.42** | -16.6 | **0.5** | **-1.1** |
| CCR1_20180203 | 14.9 | 0.20 | **0.93** | **0.95** | 11.53 | **14.91** | **5.04** | -3.56 | **6.55** | **1.54** | -11.4 | **4.2** | **2.8** |
| CCR2_20180203 | 95.0 | -0.07 | **0.52** | **0.73** | 14.26 | **38.77** | **8.43** | -9.07 | **32.76** | **0.10** | -47.8 | **29.8** | **-5.4** |
| TP1_20180322 | 36.3 | 0.39 | **0.75** | **0.83** | 18.03 | **14.26** | **10.70** | -5.30 | **2.02** | **0.77** | -13.8 | **1.2** | **1.3** |
| TP2_20180225 | 22.4 | 0.54 | **0.71** | **0.82** | 25.50 | **17.92** | **15.27** | -9.66 | **-2.62** | **-0.30** | -27.8 | **-11.5** | **-9.1** |
| TP3_20180320 | 15.3 | 0.29 | **0.31** | **0.74** | 11.40 | **11.64** | **7.91** | -2.89 | **-2.37** | **-1.49** | -6.7 | **-5.9** | **-3.5** |
| TP4_20180401 | 29.5 | 0.47 | **0.51** | **0.79** | 30.94 | **29.60** | **16.64** | -16.86 | **-16.36** | **-3.71** | -31.6 | **-29.2** | **-8.3** |
| TP5_20180307 | 42.5 | 0.47 | **0.92** | **0.92** | 30.00 | **13.80** | **13.80** | -11.76 | **7.67** | **7.67** | -36.0 | **1.0** | **1.0** |
| TP6_20180305 | 64.9 | 0.17 | **0.76** | **0.78** | 42.26 | **17.26** | **14.53** | -32.30 | **-4.49** | **4.67** | -66.1 | **-10.9** | **-2.9** |
| TP7_20180107 | 60.8 | 0.44 | **0.75** | **0.80** | 28.13 | **18.09** | **18.08** | -15.95 | **-0.48** | **6.04** | -60.0 | **-19.2** | **-12.6** |
| TP8_20180128 | 34.6 | 0.49 | **0.38** | **0.82** | 11.98 | **28.32** | **10.65** | 0.67 | **-4.91** | **2.68** | 1.1 | **-47.7** | **4.3** |
| XJ1_20180105 | 52.2 | 0.79 | **0.89** | **0.86** | 27.53 | **22.78** | **22.81** | -4.00 | **20.21** | **22.18** | -52.2 | **8.4** | **0.1** |
| XJ2_20180213 | 23.4 | 0.64 | **0.82** | **0.92** | 23.55 | **10.65** | **20.81** | 7.13 | **2.68** | **18.29** | -14.8 | **4.3** | **7.2** |
| XJ3_20180220 | 56.0 | 0.56 | **0.73** | **0.86** | 26.35 | **24.83** | **18.78** | -9.87 | **21.24** | **16.06** | -51.2 | **2.8** | **1.9** |
| XJ4_20180103 | 23.5 | 0.70 | **0.65** | **0.74** | 28.94 | **25.58** | **28.86** | 15.70 | **22.93** | **26.66** | -21.8 | **-1.1** | **-0.2** |
| XJ5_20180220 | 46.1 | 0.55 | **0.87** | **0.92** | 23.55 | **15.95** | **11.99** | -10.44 | **5.59** | **2.28** | -32.7 | **-4.1** | **-8.0** |
| Average | 40.7 | 0.46 | **0.73** | **0.84** | 23.50 | **20.02** | **14.44** | -7.51 | **7.55** | **7.17** | -33.0 | **-4.3** | **-1.9** |

Note: TAC, CGF, and STAR represent TAC NDSI, MODIS CGF NDSI, and STAR NDSI datasets, respectively.

**Response to Referee #2 Comments**

**General Comment:**

The manuscript titled "STAR NDSI collection: A cloud-free MODIS NDSI dataset (2001–2020) for China" estimates cloud-free snow data for China. The authors use Spatio-Temporal Adaptive fusion method with erroR correction (STAR) to derive snow cover. Cloud cover is the main obstacle in passive remote sensing snow monitoring and is important to overcome. The study is important but there are few major issues in the present form which needs to be addressed.

**General Response:**

Thank you very much for the critical comments and suggestions regarding our article. Considering all the constructive comments, we carefully revised the manuscript and supplemented a comparative experiment. Based on the experimental results, the priority scheme of the pre-process TAC and the accuracy of STAR NDSI collection were discussed in detail.

**In summary, the main points of the response include: (1)** Combined with the comments of referee #1, we searched various existing cloud-free snow cover products covering China and added descriptions in the Introduction. **(2)** A comparative experiment between the modified Terra and Aqua combination (TAC) and the original TAC was performed. **(3)** The uncertainty caused by the pre-process TAC and the accuracy of STAR NDSI collection were analyzed. **(4)** The proposed cloud removal method was re-emphasized. **(5)** The code was uploaded to Zenode.

**Major comments:**

Q1. The authors use combined Terra and Aqua MODIS data in this manuscript. They combine the data first and then use STAR method consisting of spatio-temporal adaptive fusion (STAF) and error correction (EC). Combining Terra and Aqua this way potentially overestimates snow (Muhammad and Thapa, 2020, 2021). The authors are suggested to either revise the TAC or explain the potential uncertainty.

**Response:** Thank you for the critical comment. Firstly, we carefully learned the cloud removal method proposed by Muhammad and Thapa (2020, 2021). Then, we analyzed the necessity of TAC using the maximum strategy. Finally, we performed a comparative experiment between the modified TAC and the original TAC to demonstrate the analysis. The priority schemes of the modified TAC and the original TAC are determined as *low*

*value > high value > cloud* and *high value > low value > cloud*, respectively.

Muhammad and Thapa (2020, 2021) suggested merging Terra and Aqua 8-day binary snow cover products in a way that considered snow only where pixels in both the products are classified as snow (i.e., *no snow > snow > cloud*). A multi-step combination method was used to remove clouds from MODIS snow cover area (SCA) products, consisting of seasonal filtering, temporal filtering, spatial filtering, and TAC. This priority scheme of TAC is an inter-verification of Terra and Aqua 8-day snow cover products which generally overestimate the snow cover extent. It also helps to avoid uncertainty produced using spatial filtering. Therefore, it is superior to the priority scheme of *snow > no snow > cloud*. However, they also suggested that "**We do not recommend this merging criteria for daily snow products** in mountainous areas due to the error of omission which may be further increased because of the off-nadir-view acquisition and edge pixels".

For STAR NDSI collection, the pre-process TAC is used to combine the Terra and Aqua MODIS daily NDSI datasets. Since there is no significant overestimation in the daily original datasets and no uncertainty caused by spatial filtering, the priority scheme is set as *high value > low value > cloud*. The four quantitative evaluations in the manuscript are shown below (Revised results based on the comments of Referee #1). Compared with NIEER AVHRR SCE (Hao et al., 2021) and MODIS CGF NDSI (Hall and Riggs, 2020) products, our STAR NDSI collection presents optimal classification accuracy and numerical accuracy. All the results reveal that our STAR NDSI collection tends to slightly underestimate rather than overestimate snow cover area. The omission errors (OEs) in Table 4 and Table 5 are relatively significant and the differences in snow rate (SRDs, MODIS – Landsat) in Table 6 and Table 7 are generally negative. These findings are in line with their assessment of daily snow products (Muhammad and Thapa, 2020). Particularly, the absolute errors (AEs, MODIS – Landsat) in the evaluations based on Landsat NDSI maps are generally positive. This is mainly due to the different spectral response curves of the bands on different sensors.

**3.1 Validation against in-situ snow depth measurements**

**Table 4. Classification statistics based on in XJ.**

| Indicators | Snow depth > 0 cm (Snow fraction = 30%) | | | Snow depth > 1 cm (Snow fraction = 28%) | | |
|---|---|---|---|---|---|---|
| | TAC | NIEER | STAR | TAC | NIEER | STAR |
| OA | 0.81 | 0.94 | **0.95** | 0.82 | 0.94 | **0.95** |
| CE | 0.02 | 0.02 | 0.04 | 0.02 | 0.03 | 0.05 |
| OE | 0.60 | 0.17 | **0.07** | 0.58 | 0.15 | **0.05** |

**Table 5. Classification statistics over TP.**

| Indicators | Snow depth > 0 cm (Snow fraction = 5%) | | | Snow depth > 1 cm (Snow fraction = 3%) | | |
|---|---|---|---|---|---|---|
| | TAC | NIEER | STAR | TAC | NIEER | STAR |
| OA | 0.94 | 0.93 | **0.95** | 0.96 | 0.94 | **0.96** |
| CE | 0.01 | 0.04 | 0.02 | 0.02 | 0.05 | 0.03 |
| OE | 0.78 | 0.53 | **0.52** | 0.72 | 0.42 | **0.39** |

Note: TAC, NIEER, and STAR represent TAC NDSI, NIEER AVHRR SCE, and STAR NDSI datasets, respectively.

**3.2 Validation based on Landsat NDSI maps**

[revised manuscript text omitted]

A comparative experiment was performed to further demonstrate the effectiveness of the original TAC, as shown in Supplementary Table 1 (not added to the manuscript). In this experiment, the priority schemes of the modified TAC and the original TAC are determined as *low value > high value > cloud* and *high value > low value > cloud*, respectively. Combining Terra and Aqua daily NDSI products with a minimum strategy, the average AE and RMSE are slightly decreased by 1.8 and 0.21, respectively. However, the SRDs are worsen compared with Landsat NDSI maps, especially on Tibetan Plateau (TP). The OEs are also expected to be worsen compared with in-situ snow depth measurements. In addition, this underestimation of snow cover area will be further amplified after the key-process STAR due to less snow cover information. As a result, we also do not recommend the pre-process TAC with a minimum strategy in the cloud removal of daily MODIS NDSI products.

**Supplementary Table 1. Performance statistics for the modified TAC (TAC_MIN) and the original TAC (TAC_MAX) in clear-sky areas according to the TAC dataset.** Note: Red bold values indicate a significant degradation of the modified TAC compared with the original TAC used for STAR NDSI collection. Purple and bold Regions/Dates are the newly added Regions/Dates on TP.

| Region_Date | CC | | RMSE | | AE | | SRD (%) | |
|---|---|---|---|---|---|---|---|---|
| | TAC_MAX | TAC_MIN | TAC_MAX | TAC_MIN | TAC_MAX | TAC_MIN | TAC_MAX | TAC_MIN |
| NC1_20180225 | 0.95 | 0.95 | 16.76 | 16.44 | 15.17 | 14.27 | 2.91 | 0.17 |
| NC2_20180311 | 0.90 | 0.90 | 12.21 | 12.20 | 11.75 | 11.74 | -0.08 | -0.08 |
| NC3_20180311 | 0.92 | 0.92 | 2.34 | 2.34 | -0.09 | -0.12 | -1.03 | -1.24 |
| NC4_20180318 | 0.98 | 0.98 | 10.34 | 9.52 | 4.90 | 4.11 | -1.20 | -1.99 |
| CCR1_20180203 | 0.83 | 0.88 | 3.35 | 2.37 | 0.71 | 0.09 | 3.67 | -0.25 |
| CCR2_20180203 | 0.54 | 0.56 | 11.00 | 10.40 | 6.90 | 6.34 | 33.43 | 31.59 |
| TP1_20180322 | 0.75 | 0.76 | 10.26 | 9.43 | 0.51 | -1.01 | **0.51** | **-4.07** |
| TP2_20180225 | 0.86 | 0.82 | 13.50 | 15.97 | 0.86 | -4.84 | **-7.51** | **-18.59** |
| TP3_20180320 | 0.75 | 0.75 | 3.69 | 3.77 | -0.40 | -0.57 | -1.10 | -1.93 |
| TP4_20180401 | 0.80 | 0.69 | 16.31 | 22.89 | -4.40 | -11.44 | **-8.42** | **-20.90** |
| TP5_20180307 | 0.94 | 0.94 | 13.57 | 12.94 | 8.21 | 6.26 | 1.83 | -1.97 |
| TP6_20180305 | 0.80 | 0.76 | 14.87 | 16.97 | 1.79 | -0.03 | **-4.75** | **-8.12** |
| TP7_20180107 | 0.98 | 0.98 | 15.13 | 14.51 | 10.27 | 9.16 | -0.93 | -3.81 |
| TP8_20180128 | 0.76 | 0.82 | 10.58 | 7.96 | 3.12 | 1.48 | 7.51 | 2.36 |
| XJ1_20180105 | 0.89 | 0.85 | 24.47 | 24.07 | 24.28 | 23.88 | 0.00 | 0.00 |
| XJ2_20180213 | 0.95 | 0.95 | 20.53 | 18.46 | 18.35 | 16.35 | 9.07 | 8.05 |

| | | | | | | | | |
|---|---|---|---|---|---|---|---|---|
| XJ3_20180220 | 0.93 | 0.91 | 17.49 | 16.21 | 15.50 | 13.90 | 4.15 | 3.74 |
| XJ4_20180103 | 0.67 | 0.65 | 29.53 | 23.54 | 28.18 | 22.23 | 0.62 | 0.56 |
| XJ5_20180220 | 0.98 | 0.97 | 8.63 | 7.75 | 3.15 | 2.46 | -1.24 | -1.79 |
| **TP9_20180125** | 0.90 | 0.84 | 10.35 | 11.96 | 0.93 | -3.29 | **-3.16** | **-12.33** |
| **TP10_20180305** | 0.83 | 0.80 | 12.97 | 14.15 | -3.44 | -5.08 | **-5.73** | **-10.40** |
| **TP11_20180107** | 0.93 | 0.94 | 12.14 | 11.70 | 4.74 | 3.93 | **-0.79** | **-5.08** |
| **TP12_20180312** | 0.97 | 0.97 | 10.24 | 8.86 | 5.11 | 2.85 | 1.13 | -0.41 |
| **TP13_20180307** | 0.94 | 0.94 | 12.12 | 12.12 | 5.30 | 4.61 | **-1.76** | **-4.63** |
| **TP14_20180226** | 0.82 | 0.81 | 12.11 | 12.76 | -4.72 | -5.41 | **-5.73** | **-8.67** |
| Average | **0.86** | **0.85** | **12.98** | **12.77** | **6.27** | **4.47** | **0.86** | **-2.39** |

Q2. The authors missed to share the code to generate STAR NDSI dataset. It is incomplete without sharing the code. The code is also required to evaluate the methodology as well.

**Response:** Thank you for the critical comment. We uploaded the code to Zenode and added a description in the Data availability as follows.

The improved cloud-free MODIS NDSI collection (STAR NDSI collection) for China from 1 August 2000 to 31 July 2020, including STAR NDSI and STAR QA data, is available for download at https://doi.org/10.5281/zenodo.5644386 (Jing et al., 2021). The dataset is provided using a WGS 84 / UTM zone 48N projection, with a tag image file format (TIFF). Users can discuss and respond to issues that arise during the use of this dataset. New versions can be released in consideration of user comments. **In addition, a source code for this collection is available at https://doi.org/10.5281/zenodo.6396149** (Jing, 2022).

Q3. The C6 snow is in NDSI ranging between 0 and 100. It is not explained how the authors reconstructed the snow data. It is a challenge to improve the data on how to replace the cloudy pixel, so it is significant to understand the way the value is replaced.

**Response:** Thank you for the comment. A Spatio-Temporal Adaptive fusion method with erroR correction (STAR) improved from our previous work (Jing et al., 2019) is utilized to eliminate cloud obscuration. This cloud removal method is detailed in the Algorithm description (Section 2.2).

**For the methodology**, the clouds are generally removed by a multi-step combination method in existing snow cover products. An accumulated cloud-free result can be obtained by combining the spatial methods and temporal methods alternately. However, the independent and successive utilization cannot take full consideration of the spatio-temporal information. Therefore, STAR NDSI collection is generated by a novel Spatio-Temporal Adaptive fusion method with erroR correction (STAR), which consists of space partition, adaptive space-time block determination, Gaussian kernel function-based fusion, and error correction. This method comprehensively

considers spatial and temporal contextual information and thus is promising for the cloud removal of NDSI products. The cloud removal procedure is described as follows.

[revised manuscript text omitted]

Q4. The authors indicate they have derived data between 2000 and 2020. The Aqua data is available from July 2002, the authors should clearly mention the observed period. As the data is combined Terra and Aqua, therefore, it should be between 2002 and 2020 not starting from the year 2000.

**Response:** Thank you for the comment. Since TAC can efficiently decrease the cloud fraction by 5%–20% with negligible precision sacrifice (Li et al., 2019), it is introduced as a pre-processing to reduce cloud coverage preliminarily. However, this pre-processing is not essential. We used STAR to completely remove clouds from Terra MODIS NDSI dataset between August 2000 and May 2002. Consequently, an improved cloud-free NDSI collection (STAR NDSI collection) for China from 1 August 2000 to 31 July 2020 can be generated. We added a clear description to the manuscript as follows.

(Lines 145-146) In addition, since the Aqua dataset is available since July 2002, the key-process STAR is directly used to remove clouds from Terra MODIS NDSI dataset between August 2000 and May 2002.

Q5. One of the major issues is the remaining overestimation. The authors have to consider the existence of overestimation mainly due to the larger solar zenith angle. It is, therefore, necessary to estimate the overestimation in the combined Terra and Aqua as in the combined product the uncertainty increases.

**Response:** Thank you for the critical comment. More rigorous evaluation and more detailed analysis were added to the Results (Section 3) and shown in the response to Q1. The related discussion is as follows:

A comparative experiment was performed to further demonstrate the effectiveness of the original TAC, as shown in Supplementary Table 1 (not added to the manuscript). In this experiment, the priority schemes of the modified TAC and the original TAC are determined as *low value > high value > cloud* and *high value > low value >*

*cloud*, respectively. Combining Terra and Aqua daily NDSI products with a minimum strategy, the average AE and RMSE are slightly decreased by 1.8 and 0.21, respectively. However, the SRDs are worsen compared with Landsat NDSI maps, especially on Tibetan Plateau (TP). The OEs are also expected to be worsen compared with in-situ snow depth measurements. In addition, this underestimation of snow cover area will be further amplified after the key-process STAR due to less snow cover information. As a result, we also do not recommend the pre-process TAC with a minimum strategy in the cloud removal of daily MODIS NDSI products.

**Overall, it can be concluded that the uncertainty caused by the pre-process TAC is slight in this study.** Undoubtedly, other approaches are likely to be valuable for different research areas and applications, such as Terra NDSI data being used alone and TAC with a minimum strategy.

**Supplementary Table 1. Performance statistics for the modified TAC (TAC_MIN) and the original TAC (TAC_MAX) in clear-sky areas according to the TAC dataset.** Note: Red bold values indicate a significant degradation of the modified TAC compared with the original TAC used for STAR NDSI collection. Purple and bold Regions/Dates are the newly added Regions/Dates on TP.

| Region_Date | CC | | RMSE | | AE | | SRD | |
|---|---|---|---|---|---|---|---|---|
| | TAC_MAX | TAC_MIN | TAC_MAX | TAC_MIN | TAC_MAX | TAC_MIN | TAC_MAX | TAC_MIN |
| NC1_20180225 | 0.95 | 0.95 | 16.76 | 16.44 | 15.17 | 14.27 | 2.91 | 0.17 |
| NC2_20180311 | 0.90 | 0.90 | 12.21 | 12.20 | 11.75 | 11.74 | -0.08 | -0.08 |
| NC3_20180311 | 0.92 | 0.92 | 2.34 | 2.34 | -0.09 | -0.12 | -1.03 | -1.24 |
| NC4_20180318 | 0.98 | 0.98 | 10.34 | 9.52 | 4.90 | 4.11 | -1.20 | -1.99 |
| CCR1_20180203 | 0.83 | 0.88 | 3.35 | 2.37 | 0.71 | 0.09 | 3.67 | -0.25 |
| CCR2_20180203 | 0.54 | 0.56 | 11.00 | 10.40 | 6.90 | 6.34 | 33.43 | 31.59 |
| TP1_20180322 | 0.75 | 0.76 | 10.26 | 9.43 | 0.51 | -1.01 | **0.51** | **-4.07** |
| TP2_20180225 | 0.86 | 0.82 | 13.50 | 15.97 | 0.86 | -4.84 | **-7.51** | **-18.59** |
| TP3_20180320 | 0.75 | 0.75 | 3.69 | 3.77 | -0.40 | -0.57 | -1.10 | -1.93 |
| TP4_20180401 | 0.80 | 0.69 | 16.31 | 22.89 | -4.40 | -11.44 | **-8.42** | **-20.90** |
| TP5_20180307 | 0.94 | 0.94 | 13.57 | 12.94 | 8.21 | 6.26 | 1.83 | -1.97 |
| TP6_20180305 | 0.80 | 0.76 | 14.87 | 16.97 | 1.79 | -0.03 | **-4.75** | **-8.12** |
| TP7_20180107 | 0.98 | 0.98 | 15.13 | 14.51 | 10.27 | 9.16 | -0.93 | -3.81 |
| TP8_20180128 | 0.76 | 0.82 | 10.58 | 7.96 | 3.12 | 1.48 | 7.51 | 2.36 |
| XJ1_20180105 | 0.89 | 0.85 | 24.47 | 24.07 | 24.28 | 23.88 | 0.00 | 0.00 |
| XJ2_20180213 | 0.95 | 0.95 | 20.53 | 18.46 | 18.35 | 16.35 | 9.07 | 8.05 |
| XJ3_20180220 | 0.93 | 0.91 | 17.49 | 16.21 | 15.50 | 13.90 | 4.15 | 3.74 |
| XJ4_20180103 | 0.67 | 0.65 | 29.53 | 23.54 | 28.18 | 22.23 | 0.62 | 0.56 |
| XJ5_20180220 | 0.98 | 0.97 | 8.63 | 7.75 | 3.15 | 2.46 | -1.24 | -1.79 |
| **TP9_20180125** | 0.90 | 0.84 | 10.35 | 11.96 | 0.93 | -3.29 | **-3.16** | **-12.33** |
| **TP10_20180305** | 0.83 | 0.80 | 12.97 | 14.15 | -3.44 | -5.08 | **-5.73** | **-10.40** |
| **TP11_20180107** | 0.93 | 0.94 | 12.14 | 11.70 | 4.74 | 3.93 | **-0.79** | **-5.08** |
| **TP12_20180312** | 0.97 | 0.97 | 10.24 | 8.86 | 5.11 | 2.85 | 1.13 | -0.41 |
| **TP13_20180307** | 0.94 | 0.94 | 12.12 | 12.12 | 5.30 | 4.61 | **-1.76** | **-4.63** |
| **TP14_20180226** | 0.82 | 0.81 | 12.11 | 12.76 | -4.72 | -5.41 | **-5.73** | **-8.67** |
| Average | **0.86** | **0.85** | **12.98** | **12.77** | **6.27** | **4.47** | **0.86** | **-2.39** |

**Minor comments:**

Q1. Line 45-65: The authors missed to point out one of the most recent cloud-free 8-day (Muhammad and Thapa, 2020 - https://doi.org/10.5194/essd-12-345-2020) and daily (Muhammad and Thapa 2021 - https://doi.org/10.5194/essd-13-767-2021) snow data, combining Terra and Aqua satellites data, reducing up to 50% of uncertainty. These datasets uniquely combine Terra and Aqua, to avoid overestimation after temporal and spatial filters are applied to individual products for clouds removal. The authors are advised to add these important papers.

**Response:** Thank you very much for the suggestion. Combined with the suggestion of Referee #1, we added and listed the existing important snow cover products in the Introduction as follows.

…On the national scale, Huang et al. (2016) obtained a long-term cloud-removed SCA product using a multi-source data fusion method. Despite many relevant studies, only a few cloud-free snow cover datasets have been released publicly.

Several typical long-term cloud-free snow cover products available online are listed in Table 1 (datasets are referenced via DOI), which cover most snow-dominated regions in China. Huang provided MODIS daily cloudless SCA products with relatively accurate snow detection capabilities in Northern Hemisphere based on multi-source data. **Muhammad and Thapa (2020, 2021) obtained eight-day/daily MODIS SCA and glacier composite datasets for High Mountain Asia by aggregating seasonal, temporal, and spatial filters, which can serve as a valuable input for hydrological and glaciological investigations.** Hao et al. (2021) yielded two long-term daily SCA datasets over China through a series of processes such as quality control, cloud detection, snow discrimination, and gap-filling (including hidden Markov random field and snow-depth interpolation techniques). Their releases and updates promoted the research of snow cover characteristics in China. Qiu et al. yielded a daily FSC dataset with detailed snow cover information over High Mountain Asia with MDC and spatial filtering. Additionally, the global cloud-gap-filled MODIS NDSI dataset (MOD10A1F) is available online since 2020, where cloud-covered grids in the MODIS Terra NDSI product are filled by retaining clear-sky observations from previous days (Hall and Riggs, 2020). However, this dataset performs poorly in China, where periodic and transient snow is dominant. In general, cloud-free SCA datasets produced by composite algorithms are frequently released, while high-quality cloud-free NDSI datasets are still scarce.

To this end, this study generates a spatiotemporally continuous Terra–Aqua MODIS NDSI product with satisfactory accuracy for China, fully considering the spatio-temporal characteristics of regional snow cover variability…

**Table 1. Typical long-term cloud-free snow cover products covering most snow-dominated regions in China.**

| References | Type | Spatial coverage | Temporal coverage | Temporal resolution | Spatial resolution | DOI |
|---|---|---|---|---|---|---|
| Hao et al. (2020) | SCA | China | 2000–2020 | Daily | ~500 m | 10.12072/ncdc.I-SNOW.db0001.2020 |
| Hao et al. (2021) | SCA | China | 1981–2019 | Daily | ~5 km | 10.11888/Snow.tpdc.271381 |
| Huang (2020) | SCA | Northern hemisphere | 2000–2015 | Daily | ~1 km | 10.12072/ncdc.CCI.db0044.2020 |
| Muhammad and Thapa (2021) | SCA | High Mountain Asia | 2002–2019 | Daily | ~500 m | 10.1594/PANGAEA.918198 |
| Qiu et al. (2017)* | FSC | High Mountain Asia | 2002–2018 | Daily | ~500 m | 10.11922/sciencedb.457 |
| Hall and Riggs (2020) | NDSI | Global coverage | 2000–present | Daily | ~500 m | 10.5067/MODIS/MOD10A1F.061 |

*Cloud coverage is less than 10%.

Q2. MODIS is onboard on Terra and Aqua satellites, the authors are advised to clearly mention which constellation they use in e.g. to clearly mention in A daily spatio-temporal continuous MODIS C6 NDSI dataset with a spatial resolution of 500 m for China (Fig. 1) from 2001 to 2020 is generated for the first time.

**Response:** Thank you very much for the suggestion. We added multiple descriptions of the constellations to the manuscript as follows.

(Lines 14-16) In this study, a recent 20-year stretch seamless **Terra–Aqua** MODIS NDSI collection in China is generated using a Spatio-Temporal Adaptive fusion method with erroR correction (STAR), which comprehensively considers spatial and temporal contextual information.

(Lines 101-102) To this end, this study generates a spatiotemporally continuous **Terra–Aqua** MODIS NDSI product with satisfactory accuracy for China, fully considering the spatio-temporal characteristics of regional snow cover variability.

(Lines 145-146) In addition, since the Aqua dataset is available since July 2002, the key-process STAR is directly used to remove clouds from **Terra** MODIS NDSI dataset between August 2000 and May 2002.

(Lines 379-380) The improved cloud-free **Terra–Aqua** MODIS NDSI collection (STAR NDSI collection) for China from 1 August 2000 to 31 July 2020.

(Lines 385-386) STAR NDSI collection is derived from **Terra–Aqua** MODIS NDSI datasets using an optimized STAR from our last research.

**Response to Referee #3 Comments**

Jing s paper produced a daily cloud-free Normalized Difference Snow Index (NDSI) product with 500 m spatial resolution based on MODIS C6 snow cover datasets in China. So far as we know, the NDSI threshold is the crucial parameter for snow detection by use of optical remote sensing data. The paper in its current version needs major revision and resubmission to meet the level expected of ESSD, for the following reasons:

Firstly, the importance of NDSI needs to be clarified in introduction and using data. NDSI is different form NDVI. The readers are more concerned about binary snow cover or fractional snow cover than NDSI itself. Therefore, it is difficult for me to evaluate whether this dataset is uniqueness or usefulness. Secondly, the current validation scheme is insufficient to support the Spatio-Temporal Adaptive fusion method. The two issues must be addressed for this dataset to be published on ESSD.

**Response:**

Thank you very much for the critical comments and suggestions regarding our article. Considering all the constructive comments, we carefully revised the Introduction and significantly improved the evaluation experiments in the Results and Discussion. Note that the NDSI_Snow_Cover (hereafter referred to as NDSI) scientific data set with a range of 0, 10 to 100 was used in this study.

**In summary, the main points of the response include: (1)** The importance of NDSI was analyzed from three aspects (Section 1). **(2)** The MODIS C6 NDSI data were compared with the atmospheric corrected NDSI calculated from the surface reflectance bands of MOD09GA products. **(3)** The overall classification accuracy and monthly average classification accuracy of STAR NDSI collection were compared with those of TAC NDSI and NIEER AVHRR SCE datasets based on in-situ snow depth measurements (Section 3.1). The accuracy of STAR NDSI collection during the snow period was emphasized. **(4)** The effectiveness of STAR method on cloud removal under different simulated cloud conditions was analyzed based on Landsat NDSI maps (Section 4). Details are presented below.

**General Comments:**

1. The importance of the NDSI research is insufficiently described, and why NDSI is more important than binary and FSC products should be further described in the introduction.

   **Response:** Thank you for the critical comment. The importance of NDSI research mainly includes three aspects: (1) NDSI is a more accurate description of the snow detection as compared to SCA and FSC (Riggs and Hall, 2015; Riggs et al., 2017); (2) As a basic data, NDSI has the significant advantage of allowing users to more

accurately determine SCA or FSC for their particular study areas and application requirements (Hall et al., 2019); (2) Cloud-free SCA and FSC datasets produced by composite algorithms are frequently released, while high-quality cloud-free NDSI datasets are still scarce. In addition, we also consider all binary SCA, FSC and NDSI products to be important. For decades, they have facilitated a variety of snow-related researches. Finally, we added a description of the importance of NDSI research in the Introduction as follows.

[revised manuscript text omitted]

variability. A Spatio-Temporal Adaptive fusion method with erroR correction (STAR) improved from our previous work (Jing et al., 2019) is utilized to eliminate cloud obscuration. The long-term detailed snow cover extent dataset will facilitate snow-related scientific studies and practical applications in China. The rest of this paper is arranged as follows. Firstly, Section 2 describes the input data and the proposed cloud removal method. Then, Section 3 presents the verification accuracy of STAR NDSI collection, with a subsequent analytical application. The cloud removal effectiveness under different cloud coverages is discussed in Section 4. Finally, the data availability and the conclusions are provided in Section 5 and Section 6, respectively.

2. The NDSI value in either MODIS C5 or C6 is the NDSI without atmospheric correction. How this NDSI differs from NDSI by the atmospheric corrected from MOD09GA/MYD09GA? Has the author compared it, and which NDSI value is more useful to readers?

**Response:** Thank you for the critical comment. The objective of this study is to improve the temporal and spatial continuity of MODIS NDSI products. The accuracy of the original data in clear-sky areas is very important but seems to be beyond the scope of this study. However, we still added a comparative experiment of MOD09GA NDSI and MOD10A1 NDSI datasets (C6). The results reveal that the accuracy of MOD10A1 NDSI dataset is slightly higher than that of MOD09GA NDSI dataset compared to Landsat NDSI maps. The reason may be that the NDSI algorithm specially designed for MOD10 products, is more applicable to them. In addition, Riggs et al. (2017) discovered that "compared to true color (bands 1, 4, 3) image of MOD09GA, all the snow-cover extent is detected in C6 NDSI_Snow_Cover by the revised algorithm; however, significant snow-cover extent was missed in C5 FSC". Therefore, MOD10A1 C6 NDSI products are relatively reliable, while MOD10A1 C5 SCA/FSC products are underestimated. Due to space limitations, this comparative experiment was not added to the manuscript. The details of the experiment are as follows.

The atmospheric corrected NDSI data were calculated from the surface reflectance bands of MOD09GA products based on the algorithm of MOD10A1 NDSI dataset (C6). Supplementary Figure 1 shows four scatter diagrams of MOD10A1 NDSI and MOD09GA NDSI datasets in snow-cover areas. The two NDSI datasets are partially inconsistent, with $R^2$ ranging from 0.63 to 0.91 and RMSEs ranging from 6.36 to 9.42. For an in-depth evaluation, the performance statistics for MOD09GA NDSI and MOD10A1 NDSI datasets against Landsat NDSI maps are shown in Supplementary Table 1. Compared with Landsat NDSI maps, their average CCs are almost equal. In addition, in terms of RMSEs, AEs and SRDs, MOD10A1 NDSI data are slightly superior to MOD09GA NDSI data. Despite the limited spatio-temporal scope of the samples, this comparative experiment can reflect the reliability of MODIS C6 NDSI products. However, more comprehensive comparisons are needed for the specific

researches on the original accuracy in clear-sky areas.

[Figure]

**Supplementary Figure 1. Scatter diagram of MOD10A1 NDSI and MOD09GA NDSI datasets in snow-cover areas.** Note that *142037_20180312* denotes region (Worldwide Reference System of Landsat) and date.

**Supplementary Table 1. Performance statistics for MOD09GA NDSI and MOD10A1 NDSI datasets against Landsat NDSI maps.**

| Region_Date | CC | | RMSE | | AE | | SRD (%) | |
|---|---|---|---|---|---|---|---|---|
| | MOD09GA NDSI | MOD10A1 NDSI | MOD09GA NDSI | MOD10A1 NDSI | MOD09GA NDSI | MOD10A1 NDSI | MOD09GA NDSI | MOD10A1 NDSI |
| 119027_20180223 | 0.98 | 0.99 | 6.29 | 5.77 | 1.65 | 1.57 | -1.67 | -1.47 |
| 119028_20180311 | 0.97 | 0.97 | 11.37 | 8.95 | 6.93 | 4.69 | -0.96 | -3.22 |
| 119029_20180311 | 0.77 | 0.81 | 2.05 | 1.80 | -0.01 | -0.02 | 0.03 | 0.04 |
| 119030_20180311 | 0.58 | 0.66 | 1.98 | 1.82 | -0.16 | -0.12 | -0.95 | -0.60 |
| 122027_20180316 | 0.91 | 0.94 | 12.34 | 9.31 | 0.18 | 0.10 | -18.36 | -12.64 |
| 123037_20180203 | 0.81 | 0.82 | 1.94 | 3.03 | -0.07 | 0.71 | -0.83 | 3.94 |
| 138039_20180401 | 0.80 | 0.83 | 15.19 | 14.60 | -2.29 | -5.80 | -6.19 | -6.62 |
| 139030_20180102 | 0.98 | 0.97 | 10.91 | 8.50 | 5.17 | 4.17 | 1.84 | 5.29 |
| 139035_20180307 | 0.91 | 0.93 | 10.65 | 7.61 | 3.85 | 1.96 | -4.07 | 2.69 |
| 139036_20180307 | 0.79 | 0.83 | 9.77 | 7.73 | 0.59 | -0.93 | -3.95 | -1.82 |
| 140035_20180226 | 0.92 | 0.94 | 8.49 | 7.75 | -0.20 | -3.13 | -2.73 | -2.36 |

| | | | | | | | |
|---|---|---|---|---|---|---|---|
| 140039_20180125 | 0.95 | 0.96 | 8.53 | 7.04 | 2.59 | 2.16 | -0.87 | 0.60 |
| 141034_20180305 | 0.87 | 0.84 | 10.57 | 11.04 | -0.37 | -4.05 | -5.30 | -7.05 |
| 141035_20180217 | 0.93 | 0.95 | 5.28 | 3.97 | 0.48 | -0.53 | -1.66 | -2.50 |
| 141035_20180305 | 0.84 | 0.87 | 11.13 | 10.55 | 2.78 | -6.12 | -3.64 | -6.00 |
| 142035_20180107 | 0.67 | 0.72 | 4.53 | 4.04 | -0.25 | -0.20 | -2.24 | -0.70 |
| 142036_20180107 | 0.91 | 0.95 | 6.63 | 4.62 | 0.15 | 0.32 | -5.75 | -0.77 |
| 142037_20180312 | 0.97 | 0.97 | 9.55 | 7.18 | 3.69 | 1.30 | -0.76 | 0.43 |
| 144028_20180105 | 0.89 | 0.89 | 24.83 | 20.75 | 24.65 | 20.58 | 0.00 | 0.00 |
| 144029_20180105 | 0.50 | 0.35 | 24.42 | 20.54 | 23.94 | 19.91 | -0.10 | 0.00 |
| 144030_20180105 | 0.65 | 0.61 | 24.59 | 21.13 | 23.10 | 19.86 | -0.61 | -0.16 |
| 144030_20180310 | 0.91 | 0.92 | 17.23 | 15.43 | 10.00 | 8.89 | 3.68 | 5.94 |
| 145028_20180213 | 0.94 | 0.94 | 20.01 | 17.72 | 16.54 | 15.52 | 4.98 | 11.33 |
| 145035_20180128 | 0.72 | 0.74 | 8.83 | 9.37 | 1.52 | 2.66 | 1.84 | 7.04 |
| 146029_20180103 | 0.79 | 0.80 | 24.77 | 21.30 | 23.89 | 20.43 | 0.14 | 0.27 |
| 146029_20180220 | 0.92 | 0.91 | 16.46 | 14.36 | 12.34 | 10.91 | 3.45 | 7.78 |
| 146030_20180103 | 0.66 | 0.64 | 26.96 | 23.61 | 25.47 | 22.32 | 0.07 | 0.69 |
| 146031_20180220 | 0.97 | 0.97 | 8.03 | 6.94 | 2.37 | 1.79 | -2.44 | -1.92 |
| 146035_20180103 | 0.82 | 0.83 | 11.37 | 12.27 | 2.66 | 5.12 | -1.10 | 9.70 |
| **Average** | **0.84** | **0.85** | **12.23** | **10.65** | **6.59** | **4.97** | **-1.66** | **0.27** |

3. The current validation plan (in-situ snow depth observations and Landsat NDSI maps) is insufficient to support the Spatio-Temporal Adaptive fusion method. Please add the improved validation plan.

**Response:** Thank you for the critical comment. We carefully redesigned the evaluation experiments and substantially revised the evaluation part in the manuscript. **In the Results**, the improved evaluation experiments included: **(1)** The overall classification accuracy and monthly average classification accuracy of STAR NDSI collection were compared with those of TAC NDSI and NIEER AVHRR SCE datasets based on in-situ snow depth measurements (Section 3.1). **(2)** The numerical accuracy of STAR NDSI collection during snow period was compared with those of TAC NDSI and MODIS CGF NDSI datasets based on Landsat NDSI maps (Section 3.2). **(3)** The evaluation in clear-sky areas and cloud-cover areas was performed based on Landsat NDSI maps, to highlight the accuracy of the recovered pixels in STAR NDSI collection during the snow period (Section 3.2). Besides, **in the Discussion**, the cloud removal effectiveness of STAR method under different simulated cloud conditions was analyzed based on Landsat NDSI maps. The new evaluation experiments were presented in the response to minor comments 7 and 8.

**Minor comments:**

1. L 95. "The daily snow cover datasets of C6 were used in this study." There are NDSI_Snow_Cover and NDSI scientific data sets in the C6 by MODIS C6 User Guide (Riggs, 2015). The NDSI_Snow_Cover and NDSI is

different, the author need to describe the data used in the study. This is related to the subsequent results.

**Response:** Thank you for the suggestion. The NDSI_Snow_Cover data were used in this study. We added a description of the scientific data set in the Data and Methods as follows.

…The NDSI_Snow_Cover (hereafter referred to as NDSI) scientific data set with a range of 0, 10 to 100 was used in this study…

2. Fig.1. It is recommended to remove the NC snow area cover. This is only an administrative division rather than a snow region (https://essd.copernicus.org/articles/13/4711/2021/). The in-situ observations of this area were not used in this study. In addition, TP suggests replacing by QTP?

**Response:** Thank you for the suggestion. Figure 1 was revised. The NC snow cover area was removed, XJ was replaced by NX, and TP was replaced by QTP (The names in the text were also revised).

[Figure]

**Figure 1. Topographic relief of China, meteorological stations in NX and QTP, and Landsat OLI scenes used for validation.**

3. What does the dashed half-frame line in Fig.2 mean?

**Response:** Thank you for the comment. The steps surrounded by the dashed half-frame line constitute our Spatio-Temporal Adaptive fusion with erroR correction (STAR) method, including spatio-temporal adaptive fusion (STAF) and error correction (EC). This is an iterative cloud removal process until no cloud remains.

[Figure]

**Figure 2. Schematic of the generation procedure of STAR NDSI collection.**

4. L 118. The description of fusion method and rules is not clear, only the priority is determined at L 123. It's better to describe the fusion method first, and then introduced the interpolation used by Aqua.

**Response:** Thank you for the suggestion. The description of TAC was revised as follows.

TAC blends the same-day snow maps deriving from MODIS sensors onboard Terra and Aqua satellites. Its cornerstone is the unlikely significant changes of the snow pattern within the data-acquired time interval (approximately 3 h). Since TAC can efficiently decrease the cloud fraction by 5%–20% with negligible precision sacrifice (Li et al., 2019), it is introduced as a pre-processing to reduce cloud coverage preliminarily. Its priority scheme is determined as high value > low value > cloud.

$$NDSI^{P} = NDSI^{Terra} \text{ IF } \left( NDSI^{Terra} > NDSI^{Aqua} \text{ OR } NDSI^{Aqua} \text{ is cloud} \right),$$
$$NDSI^{P} = NDSI^{Aqua} \text{ IF } \left( NDSI^{Aqua} > NDSI^{Terra} \text{ OR } NDSI^{Terra} \text{ is cloud} \right),$$
(1)

where $NDSI^{Terra}$ and $NDSI^{Aqua}$ are MODIS NDSI datasets from Terra and Aqua satellites, respectively. $NDSI^{P}$ represents the pre-processed NDSI maps after TAC (referred to as TAC NDSI dataset in subsequent sections). The snow in low altitude and low latitude areas during summer is reversed to no snow to alleviate commission errors inherited from the original data. In addition, since the Aqua dataset is available since July 2002, the key-process STAR is directly used to remove clouds from Terra MODIS NDSI dataset between August 2000 and May 2002. Particularly, the improved Aqua MODIS C6 NDSI dataset significantly enhances the effectiveness of TAC due to the successful restoration of the absent Aqua MODIS band 6 data by the quantitative image restoration method (Gladkova et al., 2012).

5. L158. What does NDSIP mean?

**Response:** Thank you for the comment. $NDSI^{P}$ represents the pre-processed NDSI map after the preprocess TAC. We added a description to the manuscript as follows.

…Specifically, the residual errors of the intersecting cloud-free areas of the pre-processed and fused NDSI maps ( $NDSI^{P}$ after TAC and $NDSI^{F}$ after STAF) are diffused to other cloud-free areas of the fused NDSI maps using the triangulation-based natural neighbor interpolation…

6. L 216. What does "snow-clad pixels" mean? Are there any reference?

**Response:** Thank you for the comment. To clarify the text, "snow-clad pixels" was revised to "snow-covered pixels".

7. L 214. Section 3.1 The validation method need to be improved.

The in situ snow depth derived from 49 and 92 CMA station from BJ and QTP. However, the validation date need to be clear. Due to snow-free period is long, many stations record no snow in one year. In fact, the most useful and most concerned should be the NDSI recovery during the snow cover period. The author should focus on the NDSI recovery during the snow cover period and a detailed confusion matrix needs to be given. In addition, the authors need to focus on the accuracy comparison of the product itself (TAC, L 218, the cloud-covered areas in the TAC NDSI dataset are considered to be snow-free. Here the cloud-covered areas should be eliminated without comparison) and the final spatial continuous product (STAR). The reader is concerned with the loss of NDSI accuracy after STAR interpolation.

**Response:** Thank you for the critical comment. The comment pointed to two issues, which we responded to separately.

**# Issue 1.** For the validation dates, we carefully revised the validation against in-situ snow depth measurements, including: **(1) the detailed confusion matrices were added to Table 4 and Table 5 so that the numbers of snow-cover and snow-free samples in NX and QTP during their entire validation dates are clear.** In addition, OEs ( $OE = SN/(SN + SS)$ ) also reflected the detection accuracy of snow pixels; **(2) the monthly classification accuracies in a hydrological year were added to Fig.4, and their temporal characteristics were analyzed in detail.** The corresponding revisions in the manuscript are as follows.

[revised manuscript text omitted]

**# Issue 2.** For the validation areas, all the cloud-covered areas were eliminated without comparison in the previous version of the manuscript. But the evaluation experiments were redesigned according to the comment from Chief editor Kirsten Elger. To fairly evaluate these products in the same areas, the cloud-covered areas in TAC NDSI dataset were considered to be snow-free. **On the one hand, we briefly introduced the evaluation results for cloud-free areas in the previous version**, which demonstrated that STAR NDSI collection can completely remove clouds without a significant loss of accuracy. **On the other hand, an internal comparison of STAR NDSI collection in clear-sky areas and cloud-cover areas was performed based on Landsat NDSI maps in the current version**, which highlighted the accuracy of the recovered pixels in STAR NDSI collection. The results reveal that the accuracy of recovered areas is inevitably slightly lower than that of clear-sky areas. Although the average CC decreases from 0.85 to 0.73 and the average RMSE increases from 13.48 to 16.30 compared with

clear-sky areas, the accuracy of recovered areas is satisfactory. Since many recovered areas inherit errors from clear-sky areas because the cloud removal procedure completely relies on the original dataset, a slight decrease in accuracy is reasonable. This finding highlights that STAR NDSI collection can completely remove clouds with satisfactory accuracy.

**The evaluation results for cloud-free areas in the previous version are as follows.**

**3.1 Validation against in situ snow depth measurements**

…Table 4 demonstrates that both NDSI datasets preeminently capture the snow dynamics in NX referring to the in-situ measurements, with the OAs reaching 0.97 and 0.95, respectively. CEs and OEs perform well regardless of the snow depth threshold, highlighting that remote-sensed NDSI datasets are capable of snow status estimation in NX. The generation procedure in NX has two strengths. First, the satellite-borne sensors can accurately capture the snow events on the ground due to the generally thick snow averaging approximately 20 cm. Second, the gap-filling approach with comprehensive consideration of spatial and temporal correlation has outstanding reliability due to the significant periodicity of snow variation. It can be inferred that the NDSI datasets in NC have high accuracy because of the similar snow conditions, despite the lack of in-situ data in this region.

By contrast, despite the satisfactory performance of OAs and CEs, the OEs of two NDSI datasets over QTP are as remarkably high as 28% and 39% even at the snow depth threshold of 1 cm (Table 5). This finding indicates the omission of a large number of snow-covered pixels. The specific reasons are as follows. First, the original MODIS NDSI maps frequently underestimate the snow presence throughout the snow period because discriminating the shallow snow pixels with an averaged snow depth of approximately 4 cm over QTP is challenging. Second, the credibility of the spatio-temporal contextual information is relatively low because the snow rapidly and irregularly varies due to the extremely complex topographic and climatic conditions, leading to a further decrease in the accuracy of the gap-filled results. Last, the meteorological stations over QTP are unevenly distributed and are mostly located in low- and medium-altitude/latitude areas dominated by transient snow. Consequently, the evaluation results slightly exaggerate the real OEs…

**Supplementary Table 4. Classification statistics in NX.**

| Indicators | Snow depth > 0 cm | | Snow depth > 1 cm | |
| --- | --- | --- | --- | --- |
| | TAC | STAR | TAC | STAR |
| Snow fraction | 21% | 30% | 21% | 28% |
| OA | 0.97 | 0.95 | 0.97 | 0.95 |
| CE | 0.02 | 0.04 | 0.03 | 0.05 |
| OE | 0.06 | 0.06 | 0.05 | 0.05 |

**Supplementary Table 5. Classification statistics over QTP.**

| Indicators | Snow depth > 0 cm | | Snow depth > 1 cm | |
| --- | --- | --- | --- | --- |
| | TAC | STAR | TAC | STAR |
| Snow fraction | 3% | 5% | 2% | 3% |
| OA | 0.96 | 0.95 | 0.97 | 0.96 |
| CE | 0.02 | 0.03 | 0.03 | 0.03 |
| OE | 0.45 | 0.52 | 0.28 | 0.39 |

**3.2 Validation based on Landsat NDSI maps**

…The snow dynamics presented by TAC NDSI and STAR NDSI datasets are highly consistent with Landsat NDSI maps, with an average CC of approximately 0.84. This finding highlights that STAR NDSI collection can completely remove clouds without sacrificing accuracy. The average RMSEs of TAC NDSI and STAR NDSI datasets are 13.48 and 14.64, respectively, which are mainly due to systematic overestimation (Landsat NDSI values are generally low). In terms of snow coverage, TAC NDSI and STAR NDSI datasets are respectively slightly overestimated and underestimated, with corresponding average SRDs of 0.76% and −1.48% (SRD indicates the difference of snow rate compared with the Landsat NDSI map)…

**Supplementary Table 6. Performance statistics for two MODIS NDSI datasets against Landsat NDSI maps.**

| Region_Date | CC | | RMSE | | AE | | SRD (%) | | NCR | | SRD × NCR (%) | |
|---|---|---|---|---|---|---|---|---|---|---|---|---|
| | STAR | TAC | STAR | TAC | STAR | TAC | STAR | TAC | STAR | TAC | STAR | TAC |
| NC1_20180225 | 0.87 | 0.95 | 17.10 | 16.89 | 15.07 | 15.23 | 1.02 | 2.91 | 0.73 | 0.28 | 0.75 | 0.82 |
| NC2_20180311 | 0.83 | 0.89 | 13.87 | 12.27 | 12.14 | 11.78 | −1.68 | −0.10 | 0.72 | 0.41 | −1.22 | −0.04 |
| NC3_20180311 | **0.86** | 0.92 | 8.79 | 2.31 | 0.15 | −0.09 | −2.77 | −1.02 | 0.65 | 0.43 | −1.80 | −0.43 |
| NC4_20180318 | 0.98 | 0.98 | 11.33 | 10.29 | 6.08 | 4.88 | −0.97 | −1.23 | 0.70 | 0.59 | −0.68 | −0.73 |
| CCR1_20180203 | 0.93 | 0.83 | 5.33 | 3.37 | 1.33 | 0.68 | 2.46 | 3.48 | 0.62 | 0.53 | 1.53 | 1.84 |
| CCR2_20180203 | **0.73** | 0.55 | 8.43 | 10.93 | 0.10 | 6.80 | −5.39 | 32.63 | 0.36 | 0.02 | −1.93 | 0.58 |
| QTP1_20180322 | 0.83 | 0.75 | 10.70 | 10.22 | 0.77 | 0.44 | 1.31 | 0.11 | 0.72 | 0.46 | 0.94 | 0.05 |
| QTP2_20180225 | **0.82** | 0.86 | 15.27 | 13.50 | −0.30 | 0.89 | −9.12 | −7.49 | 0.85 | 0.66 | −7.75 | −4.95 |
| QTP3_20180320 | 0.74 | 0.73 | 7.91 | 3.74 | −1.49 | −0.40 | −3.48 | −1.07 | 0.74 | 0.63 | −2.59 | −0.67 |
| QTP4_20180401 | 0.79 | 0.79 | 16.73 | 16.56 | −3.71 | −4.35 | −8.15 | −8.14 | 0.45 | 0.32 | −3.71 | −2.61 |
| QTP5_20180307 | 0.92 | 0.94 | 13.87 | 13.82 | 7.64 | 8.01 | 1.07 | 1.68 | 0.59 | 0.34 | 0.64 | 0.57 |
| QTP6_20180305 | 0.78 | 0.79 | 14.54 | 15.03 | 4.66 | 1.75 | −2.93 | −4.79 | 0.65 | 0.23 | −1.91 | −1.10 |
| QTP7_20180107 | 0.75 | 0.98 | 18.12 | 15.17 | −0.53 | 10.28 | −19.25 | −1.00 | 0.65 | 0.25 | −12.47 | −0.25 |
| QTP8_20180128 | 0.82 | 0.75 | 10.79 | 11.13 | 2.74 | 3.27 | 4.39 | 7.62 | 0.64 | 0.42 | 2.81 | 3.19 |
| NX1_20180105 | 0.85 | 0.89 | 24.87 | 24.47 | 24.67 | 24.29 | 0.07 | 0.00 | 0.93 | 0.44 | 0.07 | 0.00 |
| NX2_20180213 | 0.92 | 0.95 | 20.81 | 20.47 | 18.29 | 18.27 | 7.25 | 9.00 | 0.70 | 0.53 | 5.05 | 4.80 |
| NX3_20180220 | 0.86 | 0.93 | 18.78 | 17.45 | 16.06 | 15.37 | 1.89 | 4.02 | 0.72 | 0.32 | 1.37 | 1.28 |
| NX4_20180103 | **0.74** | 0.64 | 28.86 | 29.54 | 26.66 | 28.09 | −0.23 | 0.61 | 0.49 | 0.37 | −0.11 | 0.23 |
| NX5_20180220 | 0.92 | 0.97 | 11.99 | 8.92 | 2.28 | 3.26 | −7.96 | −1.31 | 0.69 | 0.37 | −5.51 | −0.49 |
| Average | 0.84 | 0.85 | 14.64 | 13.48 | 6.98 | 7.81 | −2.24 | 1.89 | 0.66 | 0.40 | −1.48 | 0.76 |

Note that NCR is the intersecting non-cloud rate. SRD is the difference in snow rate. Red and blue bold values respectively indicate that STAR NDSI is an improvement and degradation compared with TAC NDSI.

8.  L 248. Section 3.2. The validation method need to be improved.

The focus of validation in the study should be whether the STAR method is reliable. Therefore, a reasonable verification scheme is to select actually cloud-free the Landsat NDSI maps as a reference maps, then

artificially set a random 20%, 50% or 80% cloud cover (only my suggestion) on the corresponding MODIS data. The different cloud ratio maps were recovery after STAR interpolation and validated by reference Landsat NDSI maps, and the conclusions is convincing.

**Response:** Thank you for the critical comment. Since the validation scheme of this manuscript was designed from a product perspective, we did not include a simulated experiment in the Results. However, we also believe that the effectiveness of STAR method under different cloud conditions is important. Therefore, we added the simulated experiment to the Discussion as follows.

[revised manuscript text omitted]